# Deep learning connects DNA traces to transcription to reveal predictive features beyond enhancer–promoter contact

Aparna R. Rajpurkar [1,2], Leslie J. Mateo [2], Sedona E. Murphy [1,2] & Alistair N. Boettiger [2✉]

Chromatin architecture plays an important role in gene regulation. Recent advances in super-resolution microscopy have made it possible to measure chromatin 3D structure and transcription in thousands of single cells. However, leveraging these complex data sets with a computationally unbiased method has been challenging. Here, we present a deep learning-based approach to better understand to what degree chromatin structure relates to transcriptional state of individual cells. Furthermore, we explore methods to "unpack the black box" to determine in an unbiased manner which structural features of chromatin regulation are most important for gene expression state. We apply this approach to an Optical Reconstruction of Chromatin Architecture dataset of the Bithorax gene cluster in *Drosophila* and show it outperforms previous contact-focused methods in predicting expression state from 3D structure. We find the structural information is distributed across the domain, overlapping and extending beyond domains identified by prior genetic analyses. Individual enhancer-promoter interactions are a minor contributor to predictions of activity.

[1] Department of Genetics, Stanford University, Stanford, CA, USA. [2] Department of Developmental Biology, Stanford University, Stanford, CA, USA.
✉email: boettiger@stanford.edu

Understanding the connections between genome structure and transcriptional regulation in animal cells is of central importance to numerous biological processes. Many genes are controlled by regulatory sequences, such as enhancers, positioned thousands to millions of base pairs distal from their transcription start sites (TSS). It is widely thought that 3D genome folding allows proteins bound to distal positions to influence polymerase activity at the TSS[1–4]. However, the relative importance of enhancer–promoter proximity and other structural properties of the genome for transcriptional regulation are just beginning to be understood.

In the last decade, bulk approaches leveraging the power of high-throughput sequencing greatly expanded our understanding of 3D chromatin biology. Chromosome conformation capture (3C, Hi-C) and related methods have revealed a rich, non-random organization to animal genomes, which tend to cluster chromatin into compartments of similar epigenetic states and segregate adjacent domains into regions of increased intradomain contact, called topologically associated domains (TADs)[5–8]. These structural features have been shown to correlate with key aspects of transcriptional regulation. For example, TADs are more likely than a similarly-sized random partitioning of the genome to contain (1) co-regulated genes and (2) a gene and its distal *cis*-regulatory enhancers. In some cases, perturbations that merge TADs or replicate TAD boundaries result in ectopic gene activation or gene silencing[5–8]. However, bulk approaches measure limited aspects of structure, such as contact interactions, and observe only population level averaged structural features and averaged expression states rather than the features of individual cells.

In contrast, imaging approaches can directly measure the distances between elements, such as the position of a distal enhancer and its target promoter, in single cells[9–18]. Genetically encodable fluorescent markers have been used to estimate the distance between regulatory elements in live cells[19–26] and recent work has combined this approach with imaging nascent transcription[22,23]. Although such single-distance measurements provide limited structural information, super-resolution imaging of chromatin structure in cells[27–32] directly visualizes nanoscale structural features, allowing quantification of properties such as compaction, elongation, or the tendency to split into distinguishable globules.

Recently, new approaches for high-resolution chromosome tracing have provided a view of the 3D path of the chromatin polymer, with resolution up to several kilobases across the entire *cis*-regulatory domain of multiple genes[32–36]. Optical Reconstruction of Chromatin Architecture (ORCA)[33] Hi-M[34] and MINA[36] access this polymer information by consecutively imaging adjacent steps along the chromosome, a few kilobases at a time. Each step is visualized through hybridization of fluorescently labeled oligos, which are removed in the next step to provide sub-diffraction-limited resolution between steps[33,34]. RNA labeling provided parallel measurement of mature[33,34] or nascent transcripts from these cells[33]. As both the detailed polymer structure and transcriptional state are known in the same cells, these data provide a unique opportunity for an unbiased analysis of how higher-order structural features of chromatin relate to nascent transcriptional activity. Examples of features that can be examined using this unique data set include enhancer–promoter interactions, multi-way contact hubs, silencer interactions, and compaction, among others.

However, we currently lack the tools to leverage this data in an unbiased manner. Existing approaches used hypothesis-driven analyses, such as speculating enhancer–promoter contact activates transcription[33]. Such hypotheses require the selection of arbitrary thresholds (e.g., distance for contact), and have uncovered relatively weak correlations between structure and function[33]. The weakness of these correlations may result from either a limited dependence of gene expression on chromatin structure or the inability of the simple enhancer–promoter contact model to take into account the complexities of endogenous regulation, such as a requirement for multiple enhancers to act simultaneously on a gene to activate expression. Therefore, in order to (1) address more thoroughly to what degree chromatin structure relates to the transcriptional state of individual cells and (2) determine in an unbiased manner, which structural features of chromatin regulation are most important, we developed a deep learning-based approach, which is threshold free, and can account for a wide variety of complex structure-expression relationships.

Here, we illustrate the utility of a deep learning approach by analyzing a data set from the Bithorax Complex gene cluster (BX-C) in *Drosophila*, in which 330 kilobases of sequence control the expression of three Hox genes essential for developmental patterning[33,37]. This data set contained over 50,000 cells, in each of which the 3D structure of BX-C gene cluster was imaged with ORCA and nascent RNA expression for each of the three hox genes measured with fluorescent in situ hybridization targeting ribonucleic acid molecules (RNA FISH). This approach uncovered a significant array of structural features that augment the weak predictions provided by enhancer–promoter proximity alone. These features were distributed throughout the domain—extending further from the genes than previous genetically identified regulatory features. These features largely have redundant predictive ability, suggesting redundant layers of control. Distinct structural features were predictive of silent as well as active states, suggesting important roles for higher-order folding in gene repression.

## Results

**Neural networks predict transcription from genome structure.** In order to determine how well chromatin structure predicts gene expression (Fig. 1a), it was important to first remove features that reflected technical rather than biological differences in the chromatin structure data. For example, the $(x,y,z)$ coordinates measured for the polymer (Fig. 1a, Supp. Data 1) are recorded relative to the microscope stage axis. Viewing the same structure from two different angles results in different $(x,y,z)$ values, which do not reflect a biological difference in structure. We addressed this challenge by calculating the relative distances between all 52 positions along with the polymer, represented as a $52 \times 52$ matrix, thereby preserving all structural features except the experimental viewing angle (Fig. 1b). Missing data values were estimated by linear interpolation between the adjacent (x,y,z) coordinates. All distances were then normalized relative to the average inter-position distances to accentuate differences. To facilitate deep learning, we classified nascent RNA expression into ON or OFF binary classes (Fig. 1a).

To predict the expression state from 3D conformation, we used convolutional neural networks (CNN)[38] (Fig. 1c), a deep learning framework that has proven highly effective in image processing and other spatially structured data problems[38–40]. 2D CNNs start with an input data matrix (image), which is passed through a series of filters (convolutions). The transformed data are then passed through a data integration layer or a pooling layer. CNNs are often built with pairs of convolutional (filter) layers and pooling layers. The output of one or more rounds of convolutional followed by pooling layers is then flattened by being fed into a series of dense neural network layers, which output a final prediction. The network is optimized to maximum performance through iterative training rounds, where predictions

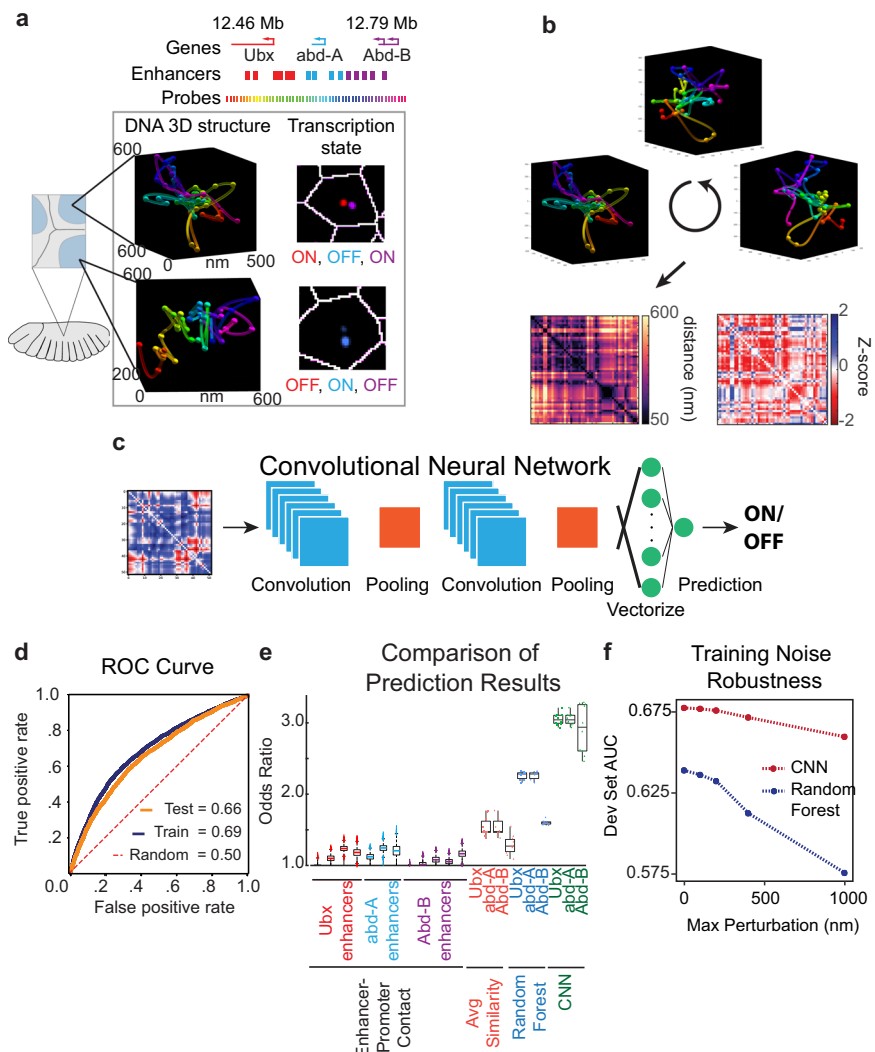

**Fig. 1 Deep learning connects transcription-state and chromatin structure. a** Schematic of major features of ORCA BX-C data set from Mateo et al. 2019. Fifty-two individual barcodes cover 312 kb. Representative images of the DNA structure are shown, where unique steps along the sequence are denoted as distinct colored balls, colors match the probe track. The line joining these balls is a guide to the eye. Example images of nascent RNA transcripts from these hox genes are also shown. **b** Viewing angle changes the absolute (x,y,z) coordinates, but not the matrix of all pairwise distances within the polymer. **c** Schematic depiction of convolutional neural network architecture. **d** Receiver operating characteristic (ROC) curve of training (blue) and test (orange) data sets for prediction of abd-A expression. The dotted red line represents the performance of random classification. **e** Comparison of odds ratios for the indicated methods of predicting transcription from the structure. Enhancer–promoter contact predictions are shown for each known enhancer corresponding to each gene. Variation was quantified by bootstrapping (*n* = 3000) resampling the data with replacement. Boxplots indicate quartiles. Whiskers extend to the furthest point within 1.5 of the interquartile range. Outliers are indicated with "+". Average similarity measures each example's similarity to the average ON and OFF structures and assigns a class label based on this measurement. Average similarity, random forest, and CNN measurements from *n* = 10-fold cross-validation are shown as dots for each trial, overlaid on the boxplots. **f** Comparison of CNN and random forest model AUC performance upon noise introduced into the training set.

are compared with the ground truth training labels, generating a loss measurement between the performance of the CNN and the maximal possible performance. Using this loss measurement, weights of the network are optimized through a method known as back propagation. This iterative process is repeated until the loss stabilizes. Additional details of network training and parameter selection are described in the Methods.

Deep learning methods can be susceptible to overfitting if appropriate care is not taken during training. This results in high predictive performance on the data used in network training but low generalizability to novel data[38–40]. Overly large networks are especially susceptible, as the network may have sufficient degrees of freedom to uniquely map all the training data, rather than finding a lower-dimensional predictive pattern in the data. To test

for overfitting a portion of the data, the validation set, is reserved to evaluate training convergence and choose training parameters, called hyperparameters. A third portion of the data, the test set is reserved only for evaluating model performance at the end of training. The overall performance of the trained model is compared across multiple stratified partitions of the data in a procedure known as K-fold cross-validation. This involves splitting the data, minus the test set, into K-fold new training and validation partitions, and re-training the model on each to account for any unknown bias in the original data split.[38–40] We utilized stratified 10-fold cross-validation to validate our model (Supplementary Data 2). We evaluated model performance using a standard approach of plotting a receiver operating characteristic curve (ROC curve). The area under the curve (AUC) value of

both the training and test ROC curves for predictions of gene expression were then compared for each gene (Fig. 1d and Supplementary Fig. 1). The substantial correspondence between these two curves indicates that the trained CNN was generalizable to new data and not overfit. Furthermore, from our 10-fold cross-validation, we see that the results are highly stable across new partitions of the data (Supplementary Data 2), indicating the model's performance is agnostic to the specific training sets used, further evidence that the CNN has not overfitted the data. We find that our CNN model is consistently capable of accurately predicting gene expression from chromatin state alone without overfitting to the training set, and with strong robustness to training set noise, and alternate train/validation partitioning.

We chose the best-performing combination of hyperparameters using the highest measured validation set score (Supplementary Fig. 2, Supplementary Data 3). To ensure the model had reached a stable plateau in training, we examined the cost over epoch curves as well as the training and validation set scores (Supplementary Fig. 3). We found that our training epoch number was well within the cost plateau with only incremental change in cost, and that this was highly stable across all cross-validations. We found that our model followed traditional behavior throughout training, and saw no evidence of a potential double descent improvement in performance[41]. In favor of selecting a maximally robust model, we decided not to utilize early stopping, but rather focus on ensuring our model was demonstrating only incremental changes the longer it trained. As a further control, we randomly shuffled the data to remove any true correlation between the structure inputs and transcription-state output, and trained the CNN on this shuffled set. This completely removed the ability of the CNN to predict transcription, as indicated by an AUC of 0.5 (Supplementary Fig. 4a). As a further control on whether the data set of ~50,000 cells was deep enough to reliably train, we explored the effects of downsampling the data on the CNN performance. Downsampling the training data (up to 50% reduction in the number of examples) resulted in only a small drop (<2%) in model performance, indicating the data set was sufficiently large to enable deep learning (Supplementary Fig. 5).

After training, we observed the CNN was able to predict transcription from structure significantly better than at random (*Ubx, abd-A, Abd-B* $p = 0.001$, Wilcoxon test over the odds ratios), indicating structural features of the chromatin domain folding are in fact predictive of gene expression, for each of the genes (*Ubx, abd-A, Abd-B*) (Fig. 1d, Supplementary Fig. 1). The prediction of expression state is not perfectly dependent on chromatin structure, as indicated by the AUC <1 (Fig. 1d and Supplementary Fig. 1), which would indicate perfect correspondence between predicted and real labels. This is expected, as many other unmeasured processes, such as transcription factor binding, also influence transcription. However, the improvement relative to a random assignment indicates just how much information is dependent on structure alone.

We then asked how the performance of this unbiased approach compared with the previous enhancer–promoter–centric approach[33]. To do this, we calculated the odds ratio of observing transcription from a given hox gene promoter when its enhancer was in proximity (150 nm) for all known enhancers. As a second comparison, we calculated the average distance matrix for all ON and OFF cells, and compared every single cell's distance matrix to these average matrices. The cell was then assigned to the class (ON or OFF) whose average it was the most similar to. This method did not require the selection of any threshold values and was not solely focused on enhancer–promoter distances.

Our deep learning approach uncovered a significantly stronger relationship between chromatin structure and expression state

than either of the comparison cases. While observing enhancer–promoter contact increased the odds of observing transcription by 0–30%, the CNN-predicted structures had 180–220% greater odds of transcription (Fig. 1e). The improvement relative to estimates derived from the average structure (Fig. 1e, green bars) indicates that this relationship depends on structural complexity in single-cell structures that are not well preserved by averaging. We concluded that there are significantly greater interdependencies of structure and expression than just enhancer–promoter proximity or a distinct ON or OFF structure.

**Comparison to alternative machine learning approaches**. We benchmarked the performance of the CNN against alternative machine learning approaches, including dense neural networks (DNNs)[39] and a random forest (RF) algorithm[42]. DNNs avoid an explicit ordering (or spatial structure) to the input data, but otherwise have a similar training approach to CNNs[39]. RFs are a popular alternative to CNNs, which construct an ensemble of decision trees to classify input data and take the mode as the prediction[42,43]. Although both approaches out-performed the predictions from pairwise enhancer–promoter contact, neither achieved the performance of the CNN assed by odds ratio (Fig. 1e). For all three genes, the performance of the CNN was significantly better (Wilcoxon test, $p < 1e\text{-}4$). The average AUC performance of the CNN was also better for predictions of all three genes ($p < 1e\text{-}4$), (Supplementary Data 2 vs. Supplementary Data 4).

We also compared the robustness of the CNN and RF approaches to added noise in the training data set. Noise in the measured positions of the chromatin trace may arise for a variety of experimental reasons, including the photon shot noise from a small number of emitters, undetected degrees of sample drift, or undetected anisotropic background signal in data collection, leading to typical uncertainty of 25–50 nm[33]. Added noise of up to 100 nm uncertainty in the training set induced a minor decrease in CNN predictive performance and a more substantial decrease in the already lower RF performance (Fig. 1f). Having observed that the CNN architecture both outperforms all other tested architectures in the odds ratio for predicting expression state from 3D structure, and is also robust to training set data noise, we examined what the CNN had learned about DNA structure.

**Blanking analysis reveals gene-specific regulatory regions**. To investigate which structural properties of chromatin were most informative to the CNN predictions, we began with a deletion-inspired approach. Similar to genetic deletion strategies that test function by replacing candidate regions with neutral non-regulatory DNA to preserve genomic spacing, we blanked genomic windows of the polymers by replacing position data within the window with the data set average value (Fig. 2). This removes any potentially informative information from this part of the polymer structure while preserving the pairwise distances among non-blanked points. The blanked test data set was then passed through the trained CNN and the performance measured by the AUC (ROC). We then converted these AUCs to a normalized predictability score, where 100 corresponds to the AUC without any data blanking (or the base AUC), and 0 represents random performance (AUC = 0.5). We explored the effect of blanking a single 6 kb step on the chromatin polymer (i.e., onemonomer) up to 30 monomers (180 kb) (Supplementary Fig. 6).

To provide an intuition for interpretation of the data blanking results, we began by analyzing simulated data. These data were produced from simple versions of two popular, distinct models of gene regulation: enhancer–promoter-loop activation of

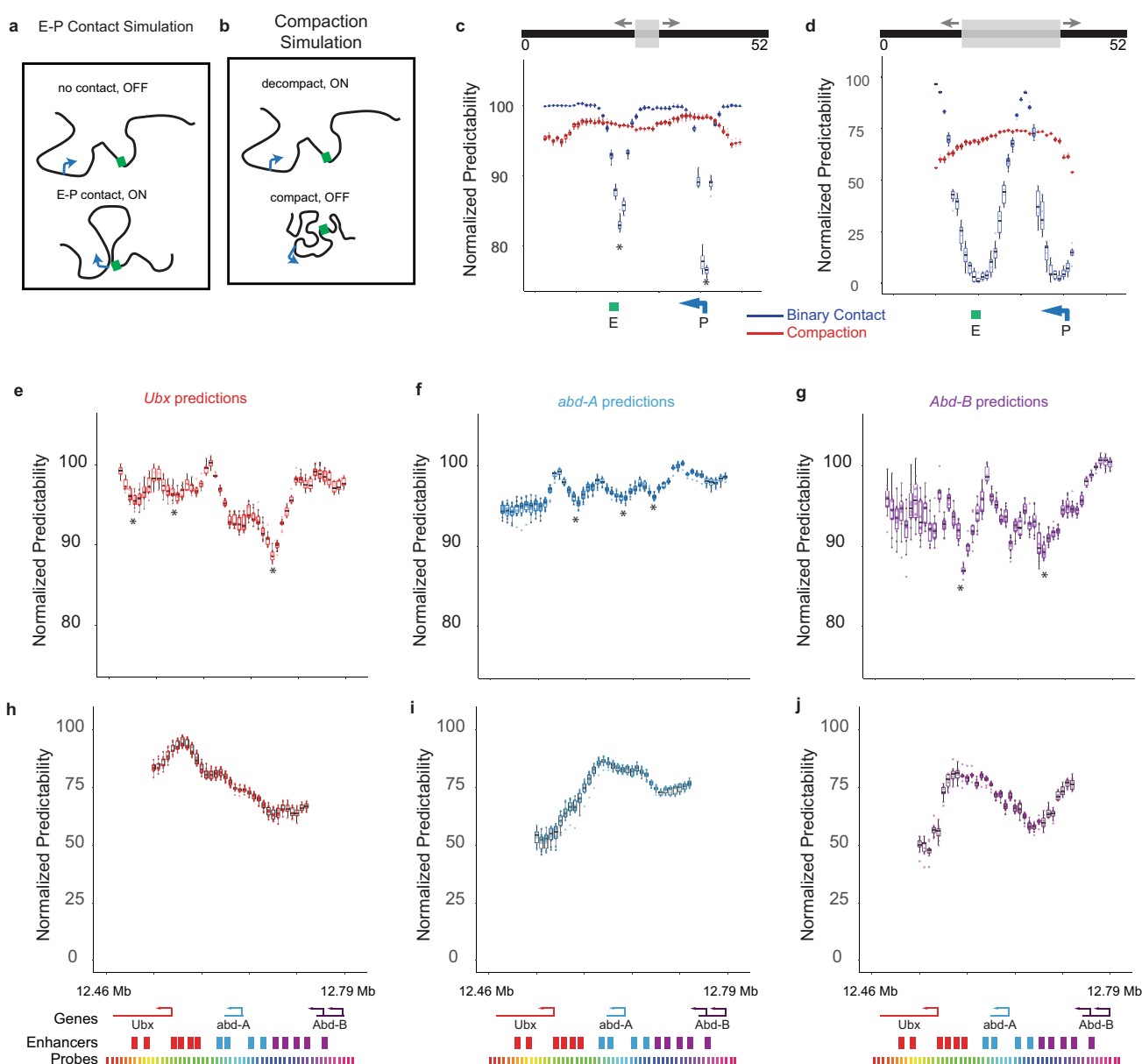

**Fig. 2 Data blanking identifies sequences driving predictions of transcription activity. a** Schematic illustration of simulation 1: enhancer–promoter (E-P) contact drives transcription. **b** Schematic of simulation 2: de-compaction drives transcription. **c** Blanking results for a window size of five monomers on two simulations, E-P contact (blue), and compaction (red). Boxplot of results shows normalized predictability, where 100 corresponds to the AUC (ROC) without any data removal and 0 corresponds to random performance. The position of the enhancer monomer 20, (green) and promoter, monomer 40 (blue) are indicated. **d** As in **c**, but with a blanking window of size 20. **e–g** As in **c**, but for CNNs predicting expression of Ubx (red), abd-A (blue), and Abd-B (purple), respectively. **h–j** as in **e–g**, but with a window of 20 steps (60 kb). In **c–h**, results from 10 independent trials used for cross-validation are shown as dots. Boxes denote quartiles. Whiskers extend to the furthest data point within 1.5 of the interquartile range. * in **e–g** indicates arbitrarily selected examples of statistically significant local minima, $p < 0.05$, relative to the average predictability, two-sided Wilcoxon rank-sum test.

transcription and chromatin compaction-mediated repression. In the loop model, polymers in which the enhancer monomer (monomer 20 in Fig. 2a) was in 3D proximity to the promoter monomer (monomer 40 in Fig. 2a) were marked ON, whereas others were marked OFF (Fig. 2a). In the compaction model, all polymers with a median inter-monomer distance below the compaction threshold were marked OFF, and those with a median distance above the threshold were marked ON (Fig. 2b). To capture the effect that chromatin structure is not the sole determinant of transcriptional state, in both cases, only 50% of the population followed the model rule and the other 50% were randomly assigned to ON or OFF. Analyses of simulations in which the contribution of the structure was more or less predictive, with cutoffs at 25% and 75% of the population following the rule, showed different absolute predictivity changes as expected but similar results in the relative importance of each sequence (Supplementary Fig. 7). Importantly, since we knew the ground truth for these simulation cases, we could directly evaluate how well the CNN could be "unpacked" to identify which properties of chromatin structure were being used for the predictions. $p$ values are Wilcoxon rank-sum test unless otherwise specified.

Examining the blanking results for the simulated enhancer–promoter loop model (E–P model) with a window of five monomers, we see no significant change in performance score ($p > 0.05$) until the blanking window intersects either the enhancer or promoter (Fig. 2c). The performance reaches a minimum when the blanking window is centered on either enhancer ($p = 6.9e\text{-}08$ for windows spanning the enhancer vs non-regulatory regions). Importantly, even though these are the only informative structural behaviors in the model, the AUC is only reduced to 85%, not 0% (Fig. 2c). This effect is a consequence of the polymer nature of the data. Unlike a random cloud of points, the remaining monomers of the polymer still provide a constraint on the likely position of the blanked monomers. Upon increasing the blanking window to size 20, we found a similar trend—the largest drop was centered at the two structurally relevant loop elements (Fig. 2d). This much larger window completely removed the ability of the CNN to infer loop interactions for windows centered on these elements, where performance hits 0% ($p > 0.7$ for a difference from 0).

In contrast to the behavior of the loop model, we find an effect at all positions in blanking data from the compaction simulation (Fig. 2c), ($p = 9.8e\text{-}04$ for all positions compared with non-regulatory regions from the E-P model). Thus, the analysis uncovered the distributed nature of regulation by compaction—all parts of the domain contribute information to determining the degree of compaction. Moreover, the effect is notably smaller than for the E-P model—the information is not contained in any single contact, but is in effect redundantly distributed across the domain. The drop is similar across all windows, save for those near the ends of the polymer. This edge effect is likely owing to CNN's ability to infer the position of missing monomers from their neighbors. For the edge monomers, the CNN would have less data to constrain this inference. Blanking larger windows further decreases model performance ($p = 1.9e\text{-}129$ window size 5 vs 20), but in no case drops performance to 0%. Even removing 2/5th of the domain, performance remains over 50% (Fig. 2d), as the remaining monomers still contain information about the overall compaction of the polymer. These two simulation examples provide a useful source of comparison as we interpret the experimental data.

In the experimental data, we observe more-complex patterns than in the simulations. Using small blanking windows, we found that all three genes show evidence of distributed regulation. All blanking windows cause a drop-in predictability (for all but the few positions indicated, this was statistically significant $p < 0.05$) (Fig. 2e–g), unlike in the binary contact simulation where only windows overlapping the critical positions showed a drop. In addition, we observe multiple local minima throughout the domain, ($p < 0.01$ for indicated minima (*) compared with the median of all domains), indicating the existence of distinct, *cis*-regulatory elements whose physical position is informative of transcription (such as enhancer–promoter proximity). No single element has as large an effect as in the loop model, suggesting redundancy in this regulation. Interestingly, these local minima occur both inside and outside the positions of the previously identified enhancers for each BX-C gene. Finally, the three genes each had substantially distinct sensitivities to domain blanking ($p < 3.0e\text{-}8$ K-S-test), indicating distinct structural influences on their expression.

Using large windows, we still observed no complete loss of predictivity (minimum performance >40%), contrasting the enhancer–promoter simulation results and supporting the observation for distributed regulation (compare Fig. 2h–j and Fig. 2d). The widespread effects and general sensitivity of end positions are similar to the effect seen in compaction and suggestive that compaction does contribute to regulation.

However, the predictions were not always symmetrically most sensitive at the ends of the domain, contrasting a mechanism of regulation based purely on compaction. Interestingly, for all three genes the predictions were significantly sensitive to blanking windows distal to the transcription unit, which extends to the other gene bodies (Fig. 2e–j). This suggests a prominent role for long-range regulation and physical interaction between the genes in controlling transcription.

Together, these analyses indicate that much of the structural information the CNN has used to predict transcription activity is physically distributed across the domain, spanning 10 s to 100 s of kilobases of neighboring chromatin, where no individual element accounts for >20% of the total effect. Notably, although these predictive regions overlap domains identified by prior genetic analyses[37,44,45] they also extend beyond the most distal known enhancers of these genes.

## SHAP analysis identifies distinct features that predict activation and silencing.

Although an effective way to identify the position and importance of different structural features of the domain in predicting transcription activity, the blanking analysis does not tell us whether the features used were more important for predicting the active or silent state. Furthermore, blanking only indicates if the position of a certain region is important. It does not identify if proximity or separation between such regions is predictive. To answer these questions and generate mechanistic hypotheses linking structure to expression, we developed a complementary approach to distinguish ON and OFF-predictive events and determine whether contact or separation was most important.

SHAP (SHapley Additive exPlanations)[46] is an approach commonly used to interpret machine learning models and understand what they have learned[46–50]. The SHAP value is the estimated difference between the expected feature importance (based on a background reference data set) and the actual feature importance (see Methods). The major benefits of using SHAP as a method to open the black box are the dual mathematical guarantees of global and local interpretability[46]. Unlike many other feature importance methods, SHAP values are assigned on a per-example basis, such that each example can be assigned its own set of values, and the collective aggregation of these SHAP values for any one feature show the overall magnitude and directionality of that feature's contribution to predictions across the data set[46]. Because of these unique properties, the lack of need to perturb either the model or the input data externally to produce a readout of feature importance, and their ability to showcase feature combinations and interactions, SHAP values are one of the most-used approaches to understanding deep learning models.

In our CNN models, a positive SHAP value indicates that a feature influenced the model to assign the example to the ON class, whereas a negative SHAP value indicates that a feature was influential in the assignment of that example to the OFF class. Observing the SHAP values of the pairwise distance maps of individual examples, therefore, gives an understanding of which specific distances the CNN was most dependent on to make the correct classification of that example. Any individual structure can have both positive and negative SHAP values. Where the SHAP value map shows that interactions in a particular chromosome were indicative of an ON or OFF transcriptional state, the corresponding distance map for that chromosome reveals whether these interactions corresponded to contact, proximity, or separation (Fig. 3a).

We began by analyzing simulated data to assess how well this method could disentangle complex regulatory schemes with a

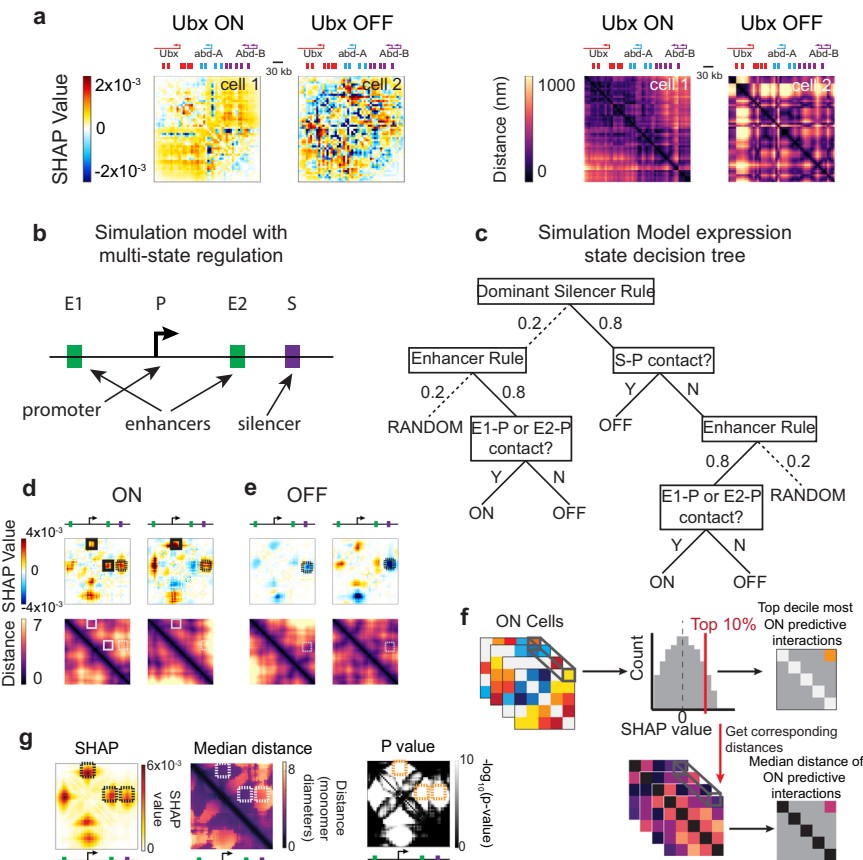

**Fig. 3 SHAP analysis interpretation on complex simulation. a** Examples of paired SHAP value maps and paired distance matrices from single cells in the ORCA data set. **b** Cartoon of simulation element structure. **c** Schematic showing hierarchical rules of the simulation. **d** Examples of paired SHAP value maps and distance matrices for ON cells. Columns represent individual examples, rows represent SHAP value maps and distance maps, respectively. Schematics above columns represent positions of two enhancers (green), promoter (black arrow), and silencer (purple). **e** Examples of paired SHAP value maps and distance matrices for OFF cells. **f** Schematic explaining population-level SHAP analysis. **g** Population-level top decile SHAP values for ON simulation examples, associated median distances, and associated p values. Colored boxes mark regions of the plots discussed specifically in the text.

known ground truth before applying it to the real BX-C data. Accordingly, we designed a simulation in which multiple enhancers (E1 and E2) and one silencer (S) function in a contact-dependent manner to regulate expression from a single promoter (P), where the silencer is dominant over the enhancers (Fig. 3b, c). We simulated a large number of polymers using molecular dynamics as before and classified them as ON or OFF with the above set of hierarchically structured rules, including a stochastic component to take into account the effects of TF binding/unbinding and potential temporal lag between expression and structure (Fig. 3c). This produced 50,000 individual cells with polymer structures and expression states following the known rules for subsequent SHAP analysis.

Figure 3d shows two examples of SHAP maps from simulated ON cells. The map values are primarily close to zero, indicating most interactions are not informative of the transcriptional state, consistent with the ground truth for the model. On the left map, at the coordinate representing the interaction between E1 and P and between E2 and P (black solid boxes), large positive values show that the distance between these elements is indicative of transcription. The corresponding distance map (Fig. 3d) shows these informative distances (white solid boxes) have low values confirming that it is E-P proximity that leads to the ON prediction (and not just that E-P distances are important). In the right map of Fig. 3d, only a single E-P distance has positive SHAP values and once again we see it corresponds to contact in the pairwise distance map (solid boxes). Weakly positive SHAP

values are assigned to the S-P interaction, which the distance map shows are far apart in this cell (dotted boxes). The silencer contributes less to the ON prediction since many cells that lack S-P contact are still OFF as they also lack E-P contact. For example in the OFF cells, we find that it is the enhancer–silencer interaction that is marked as most predictive (Fig. 3e, dotted black box) and that accordingly, corresponds to the proximity between the S and P positions in the distance map (Fig. 3e, dotted white). In the second (right) OFF example, the E-P SHAP values are blue, indicating that these are counter-indicative of the OFF state. The corresponding distance map shows these enhancers are in proximity to the promoter, which would suggest an ON state. However, the CNN has correctly learned the hierarchical relationship that while E-P contact indicates ON, it is overridden by the S-P contact for a final prediction of OFF.

Thus, SHAP analysis on select examples shows signatures of the underlying regulatory rules. Traditionally, SHAP is used to understand the interpretation of individual images[46,51,52]. Through repeated anecdotes, it is then inferred what features CNN has "learned" about the image. For example, SHAP analysis of images of cats that repeatedly highlight the ears may be interpreted as the CNN "has learned to recognize cats by their ears". However, this anecdotal approach makes it difficult to derive general, population-level conclusions. This is especially true for heterogeneous input data in which diverse features (structures) may indicate the same classification (ON/OFF), as seen in examples from our simulated data (Fig. 3d, e) and

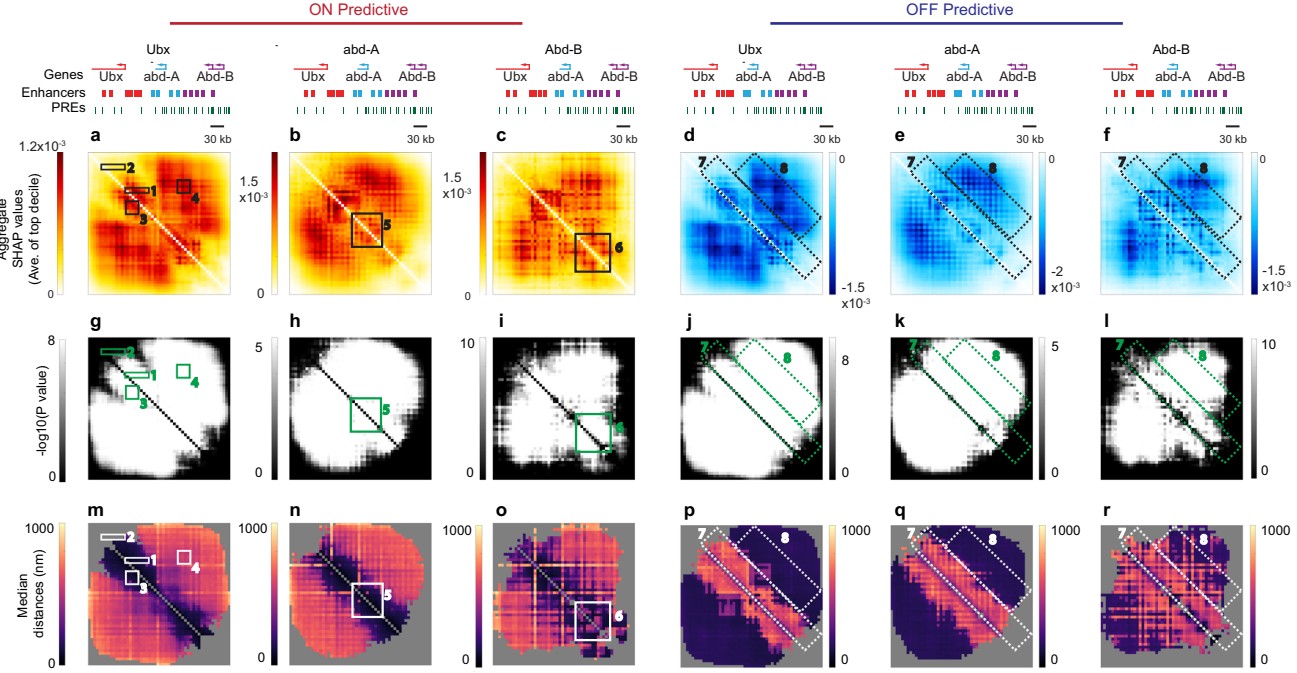

**Fig. 4 Integrated SHAP analysis identifies distinct elements associated with activation or repression. a–c** Top decile of SHAP values for all pairwise distances from all ON cells for each gene model *Ubx, abd-A, Abd-B*. Selected regions of interest highlighted in numbered boxes 1–6. **d–f** Bottom decile of SHAP values for all pairwise distances from all OFF cells for each gene model, in parallel to **a–c**. **g–l** −log10(*P* value) maps of SHAP values (**a–f**). **m–o** Median distance among the top decile of SHAP values in **a–c**. **p–r** Median distance among the bottom decile of SHAP values **d–f**. Numbered boxes mark regions of interest discussed in the text.

experimental data (Fig. 3a, b). However, given the unique organization of the distance maps in our study, we are able to develop a statistical approach to aggregate SHAP values, providing an unbiased and global view of the key features rather than an anecdotal one.

To achieve this, we created population-level SHAP maps by plotting a new map of the highest SHAP value obtained across any ON cell for each interaction in the matrix (Fig. 3f). This shows for any interaction, however rare, just how predictive it is of the ON state when it is at its most predictive. This would allow features that occur rarely in the population but which are very predictive of the ON state to still be emphasized. It also avoids canceling effects that could arise averaging large positive values with large negative ones. To avoid bias from limiting sampling, we found the average of the top decile, instead of the maximum, gave similar results more robust to sample size (Supplementary Fig. 8). To determine the associated distances at this population-level view, we examined a second matrix that plots the median distance among interacting pairs whose interaction was in the top decile of SHAP values. A similar approach was used among OFF cells, looking instead at the decile of most-negative SHAP values, to determine features predictive of the OFF state (Fig. 3f, Supplementary Fig. 9).

When applied to the simulation data, we see this approach correctly identifies the contact-dependent enhancer–promoter activation. It also uncovered the dominant, contact-dependent silencing interactions built into the model (Fig. 3g, dashed boxes, Supplementary Fig. 9). A map of the *p* values for which positions in the SHAP maps are statistically significantly also highlights the enhancer–promoter and silencer promoter interactions as the regions of most significance (Fig. 3g). Similar results were achieved with additional simulations in which the relative distance between enhancer and promoter was altered (Supplementary Fig. 10), or when different cutoffs for the contribution of the structure were used (Supplementary Fig. 11).

**SHAP de novo identifies enhancers and silencers.** Turning to *Ubx*, we find a substantial portion of the map has SHAP values statistically distinct from zero, contrasting the simple model (compare Figs. 3g and 4a). There are notably more local maxima in the integrated SHAP map than observed in the simulated example, suggesting the regulation is more distributed. Several hot spot areas stand out. For example, the ~60 kb domain spanning the *Ubx* promoter and its cluster of upstream enhancers (box 1) shows significantly higher SHAP values ($p < 1e-20$, K-S two-sample test) than a corresponding sized region (box 2) inside the gene body (Fig. 4a). The corresponding position on the distance map shows this structural feature was predictive when its distances were small, <100 nm (Fig. 4a, box 1). This supports a contact/proximity-dependent mechanism for enhancer function. This effect is statistically significant (($p < 1e-20$, predictive vs non-predictive, one-sided Wilcoxon rank-sum test). Consistent with this finding, these distances are also highlighted in the OFF-predictive maps, where larger values indicative of physical separation between enhancers and promoters is predictive of the non-transcribing state (Fig. 4d), ($p < 1e-20$, predictive vs. non-predictive, one-sided Wilcoxon rank-sum test). Even though the CNN did not know which positions contain promoters and enhancers, it recovered the experimentally validated link between the Ubx promoter and these enhancers. Moreover, unlike the blanking analysis, the SHAP analysis shows this interaction is proximity-dependent, as hypothesized from prior experimental analysis[44,45]. This example illustrates the ability of this SHAP approach to unpack biologically interesting interactions.

Several other structural features, however, were similarly informative. For example, enhancer–enhancer interactions upstream enhancers of *Ubx* have similarly high SHAP values ($p < 1e-5$ for the difference from zero) as the enhancer–promoter interactions (Fig. 4a box 3). This could reflect enhancer–enhancer communication that is predictive of transcription, or a physical interaction resulting from enhancers preferentially scanning this

domain when active, as hypothesized previously[33]. In addition, significantly slightly elevated SHAP values extend downstream of the promoter to the intronic enhancers in a contact-dependent fashion in both maps (Fig. 4a). These intronic enhancers are necessary to maintain the lower levels of *Ubx* observed in the posterior-thoracic body segment, a small subset of cells in the data[44,45].

Long-range interactions between the *Ubx* control regions and the region around *abd-A* and downstream of *Abd-B* all show statistically elevated SHAP values in both ON and OFF-predictive maps (Fig. 4a, box 4). In the ON case, these values are associated with large distances and in the OFF case with small distances (Fig. 4a), similar to the contact-dependent silencer element in our simulation (Fig. 3g). Interestingly, the degree of asymmetry in the strength of the SHAP values for the contact-dependent silencing in the simulation is not recapitulated in the data. This would arise if the absence of the repressive contact is alone permissive of transcription in some cells without enhancer–promoter activation. Intriguingly, these long-range contacts predictive of repression involve two regions rich in polycomb response elements (PREs). Classically, PREs are known to maintain local repression[53,54], though some evidence suggests long-range PRE–PRE interactions may facilitate stronger silencing[55–58].

Similar patterns of domains of enhancer–promoter and enhancer–enhancer proximity are observed in the predictive patterns for *abd-A* and *Abd-B* (Fig. 4b, c), for example, the domain spanning the *abd-A* gene and its enhancers (Fig. 4b, box 5) or the downstream regulatory region of *Abd-B* (Fig. 4c, box 6). Although it is general proximity among regulatory elements that the network finds predictive, the *Abd-B* regulatory locus provides a more-complex picture where separation of some elements is favored (Fig. 4c). This may reflect the altered regulatory structure where, in some *Abd-B* expressing cell types, more proximal regulatory elements remain silent and must be bypassed by more distal elements to activate the promoter[44]. There is some significant elevation of SHAP values around the genes that are not being predicted. As a significant number of cells in posterior body segments transcribe one or more of these genes at once, whereas the anterior segments repress all three, these values may reflect the network has learned to factor in this co-expression in prediction rather than a *cis*-regulatory interaction, an indirect effect of cell type. They may also reflect *cis*-regulatory interactions between the regulatory domains of each gene that were masked in previous genetic studies by the effects on the primary target genes.

The integrated distance maps associated with the OFF state share an interesting pattern for all three genes. Sequences within ~50 kb of one another (box 7) typically show notable separation (>500 nm) when their relative distances are at their most predictive of the OFF state (Fig. 4d–f). Sequences 50–200 kb (box 8) apart typically show close proximity (<200 nm) when they are at their most predictive values. This latter observation, proximity of distal elements, indicate compaction of the domain, a structural feature previously observed for *Pc*-repressed chromatin[9,10,28–30,59]. This observation indicates the network has identified compaction as a repressive feature. However, the form of compaction that the CNN finds most predictive is unusual, in that relatively large distances (>500 nm) are preferred among proximal (<50 kb) elements at the same time that small distances are preferred among distal elements ($p < 1e-20$, one-sided Wilcoxon rank-sum test, box 7 vs. box 8, for each *Ubx, abd-A, Abd-B*) (Fig. 4d). This surprising inverted architecture suggests a solution for the paradox of separating enhancers and promoters while compacting the overall domain. Such unusual compaction also places significant constraints on the molecular mechanisms which achieve it. For example, it is inconsistent with textbook depictions of heterochromatic compaction into 30-nm or larger

organized fibers[60], and is generally more consistent with a polymer in a confined volume[61].

**Higher-order chromatin interactions inform CNN transcription predictions**. Next, we investigated if the CNN learned about more-than-pairwise interactions that are predictive of transcription, such as the formation of enhancer hubs. To benchmark the ability of our approaches to unpack this information from the CNN, we started by comparing results from two simulations that differed only in the cooperativity among contacts. For a non-cooperative example, we considered the model with two enhancers introduced in Fig. 3a. We compared this with a modified version in which individual enhancer–promoter contacts had only a minor bias to the ON state (10%) but simultaneous contact of both enhancers had a substantial bias (80%), simulating a cooperative effect from hub formation.

To quantify the degree of cooperativity learned by the model, we computed for every pair of genomic positions $(a, b)$ that had high SHAP values in an individual ON cell (top percentile), the frequency that any other pair $(c, d) \sim= (a, b)$ also had high SHAP values (top percentile). In simulated data produced under the cooperative model, we observe, for interactions between E1 and P, $(a = 5, b = 20)$ this produces a map with a clear punctum connecting P and E2 $(a = 20, b = 35)$, but no peak in the non-cooperative case (Supplementary Fig. 12a). We computed such interaction maps across all possible combinations of four points (Supplementary Fig. 12b). We summarized this large interaction space by averaging over all possible $(a, b)$, creating a simpler map of which pairs exhibit cooperativity with any other pairs (Supplementary Fig. 12c, Fig. 5a–f). In data from the cooperative model, reveals that both E1–P and E2-P have higher-order/cooperative interactions with another position (Fig. 5a). In the data from the independent enhancer simulation, we observe much weaker peaks (Fig. 5d). The peaks are non-zero owing to the uncorrected polymer effects—when E1–P is predictive, the distances between more distant parts of the polymer are also predictive, though weaker. The cooperative interactions are statistically signficant at and around the E–P contacts ($p < 0.01$) and not particularly significant for interactions that do not involve the *cis*-regulatory elements or their immediate neighbors ($p > 0.01$) (Fig. 5b). The map of the inter-element distances associated with these cooperative interactions has small median distances (Fig. 5c), indicating that these are cooperative contacts. Notably, if we examine the highly negative SHAP values among the OFF cells, we see the opposite result. Now the independent enhancer model shows cooperativity (as the enhancers must both be disengaged from the promoter in the same cell to be likely OFF), and the corresponding distance map confirms this cooperative effect is for separation (Fig. 5g–l).

We next applied this approach for detecting the contribution of hub-like interactions to predict transcription to our experimental data (Fig. 6a–r). In the case of *Ubx*, much of the map shows evidence of statistically significant cooperative interaction, though the strength of such interaction for any position is weaker than that used in our model simulation (Fig. 5a vs Fig. 6a). This broad cooperativity indicates that higher-order contacts play a significant role in the CNN's performance, beyond the simple pairwise associations discussed in Fig. 4. For example, we found a modest degree of cooperativity among upstream enhancers (Fig. 6a, box 1), which is significant ($p < 0.01$) and corresponds to the physical proximity of the enhancer and promoter region (<150 nm) (Fig. 6g, m, box 1). No significant cooperativity was detected among the intronic enhancers (Fig. 6a, box 2). The regions with the strongest cooperative effects are distances between the entire *Ubx* regulatory unit and the regulatory regions

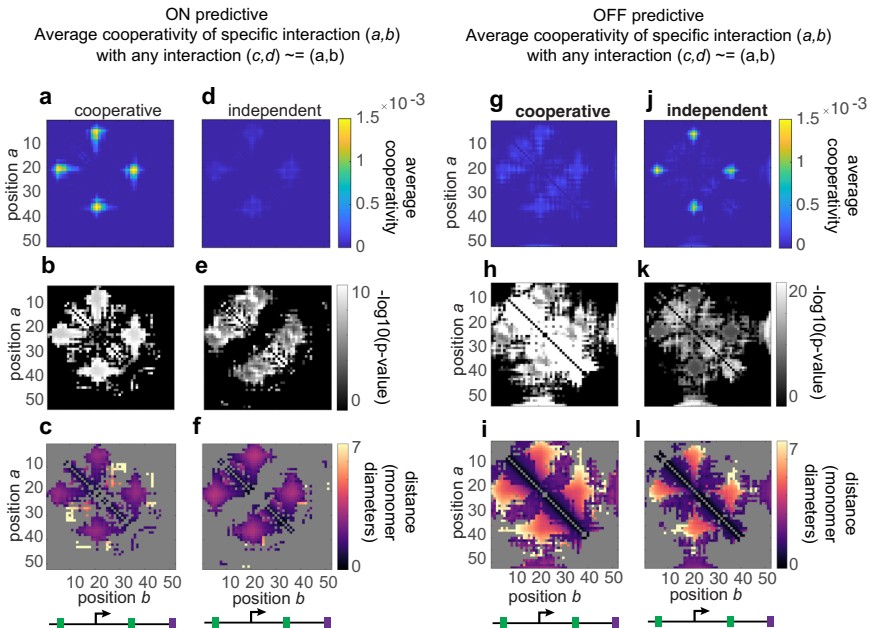

**Fig. 5 Correlation analysis of simulated models with cooperative enhancers. a–c** Correlation analysis of cooperative simulation, ON cells, consisting of **a** average cooperativity map **b** −log10(*p* value) map, and **c** corresponding median distance map for cooperative interactions. **d–f** as in **a–c**, for independent, non-cooperative simulation. **g–i** as in **a–c** for OFF cells in cooperative simulation. **j–l** as in **a–c** for independent simulation.

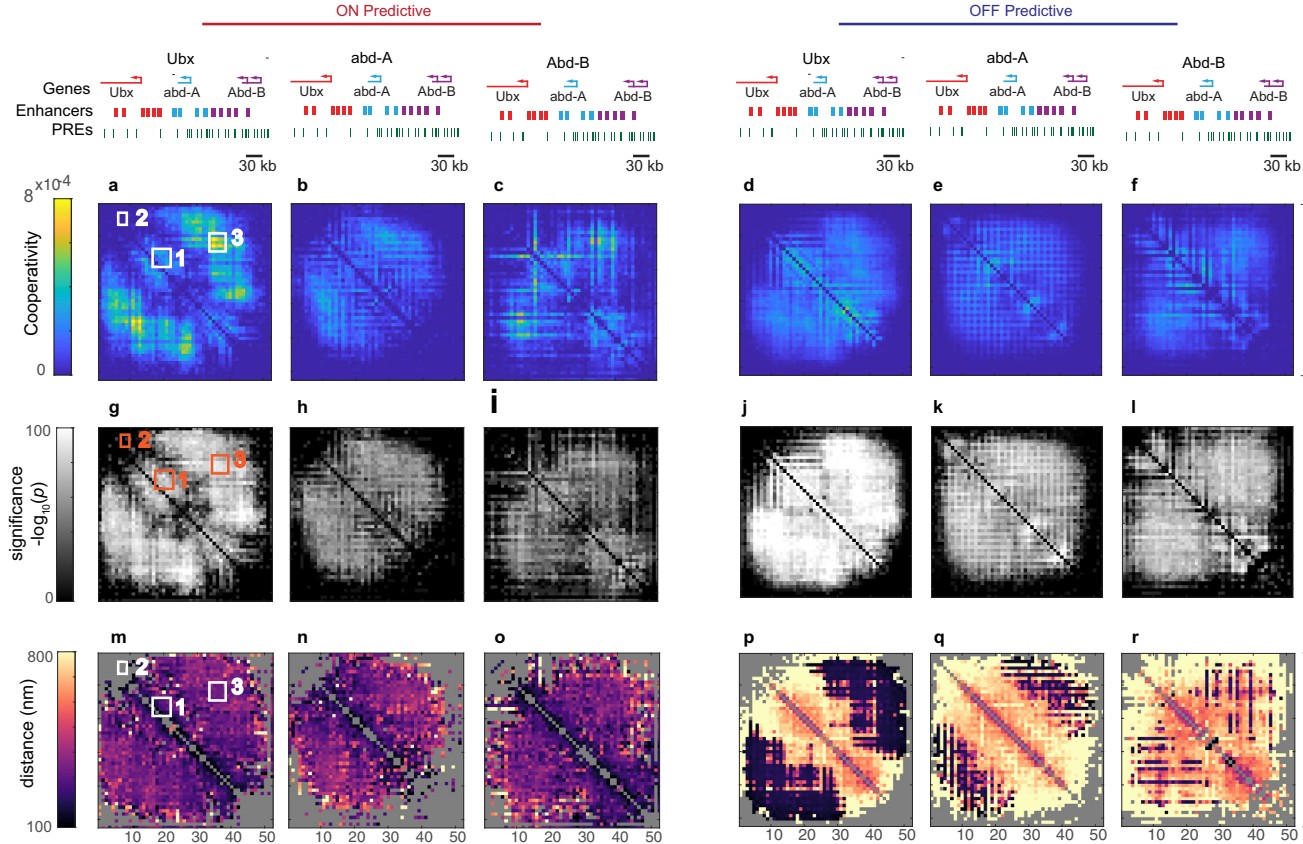

**Fig. 6 Cooperativity analysis on genes from the BX-C.** Similar analysis as in Fig. 5., on each gene model: *Ubx, abd-A, Abd-B*. **a–c** average cooperativity map on highest decile SHAP results, −log10(*P* value), and corresponding median distance of cooperative contacts for ON cells per gene: **a** *Ubx*, **b** *abd-A*, **c** *Abd-B*. **d–f** as in **a–c**, but for the lowest decile SHAP results in OFF cells for each gene. **g–l** −log10(*p* value) significance maps for cooperativity plots in **a–f**. **m–r** Median distance maps corresponding to **a–f**. Numbered boxes mark regions of the maps discussed in the text.

of *abd-A* and *Abd-B* (Fig. 6a, g, m box 3). These distances are substantial (>300 nm), indicating a simultaneous physical separation of the *Ubx* regulatory region from the control regions of *abd-A* and *Abd-B* is predictive of the active transcription of *Ubx*. For predicting the OFF state, a moderate dependence on more-than-pairwise distances is seen throughout the map, with few hotspots (Fig. 6d, j, p). The distance map reveals opposite cooperative behaviors for linearly proximal vs. distal elements. Enhancers and other proximal sequences are simultaneously far from the promoter in the OFF state (Fig. 6d–r). Meanwhile, long-range (in terms of linear sequence) interactions show the most cooperativity when they are close in 3D—indicating general compaction of the domain and not just isolated long-range contacts with potential cis-regulatory silencers (Fig. 6d, p). Close-range interactions show the most cooperativity when they are far apart in 3D from one another (Fig. 6d, p), emphasizing the inverted structure of repressed chromatin discussed above. Both *abd-A* and *Abd-B* share this general pattern of cooperative, moderate distance separation among long-range interactions in ON cells (Fig. 6b, c, n, o), the proximity of long-range elements in OFF cells, and separation of close-range elements in OFF cells (Fig. 6e, f, q, r).

## Discussion

Here we have introduced a deep learning approach for the analysis of paired super-resolution imaging of chromatin and nascent transcription data. Our approach is not dependent on prior assumptions about how chromatin structure influences transcription or reliant on arbitrary thresholds, in contrast to many previous approaches. Importantly, the method is capable of teasing out combinatorial regulatory mechanisms and hierarchical relations in this regulation, as validated with our use of simulated regulatory architectures.

Although a powerful way to identify previously unappreciated predictive features in the data, machine learning approaches and CNNs have several notable limitations. The degree to which we can unpack what the approach has learned remains incomplete, despite recent improvements in this area such as the SHAP approach we have adapted here. Compared with the mathematical descriptions of laws of physics, for example, it is difficult to build similarly deep intuition from a well trained machine learning algorithm. Most importantly, where a mechanistic mathematical model makes clear assertion of causation; the useful features, identified by unpacking the CNN, do not distinguish purely correlative from causative inputs. Nonetheless, it can highlight features in the data previously unappreciated, and we are excited to test experimentally the transcriptional importance of several structural features identified here.

Applied to the available data, the method has identified multifaceted aspects of chromatin structure that are predictive of transcription. Many of these predictive features receive little attention in existing models or hypothesis-driven investigations of animal gene regulation, which tend to emphasize single master control elements, such as super-enhancers and locus-control regions. We found structural features that predict active transcription in the BX-C domain are highly spatially distributed. Although enhancer–promoter proximity did emerge from the unbiased approach as predictive of transcription activity, we found no single element is especially informative. In contrast to recent speculation of enhancer–enhancer interactions forming a cooperative hub[62–64], the predictive accuracy of the CNN did not depend appreciably on cooperative interactions. We speculate that other genes regulated by enhancer clusters (also called super-enhancers) may also function through independent rather than synergistic interactions. In addition, CNN found that *cis*-element

proximity is important in the BX-C, contrasting recent suggestions, based on analysis of gene loci distinct from these, that *cis*-interactions may occur without change in proximity between regulatory elements[23,65]. These differences may reflect different classes of *cis*-interaction that are proximity-dependent or independent.

Although the majority of discussion surrounding chromatin structure's effect on developmental gene expression has focused on transcription activation, our approach also identified diverse structural features in the BX-C predictive of silencing. We speculate that some of these features may represent structural mechanisms of repression of relevance to genes beyond the BX-C as well. These features included a compact state, previously associated with Polycomb repression[66,67] a surprising inverted domain architecture, and specific long-range *cis*-regulatory contacts. The inverted organization is reminiscent of a polymer confined within a volume of diameter much smaller than the polymer length with sufficient time to equilibrate[61] and consistent with prior sub-domain analysis of Pc-repressed chromatin[28]. This organization may reflect the mechanisms of repression and is consistent with the formation of a spatially segregated compartment. Long-range (>5 kb) *cis*-regulatory contacts associated with repression, like those identified by the CNN, have been largely ignored in our understanding of development, though emerging work suggests they are more common and important than previously appreciated[68]. Notably, the repressive interactions identified by the CNN connect PRE-rich sequences. Although PREs are largely thought to function in a local manner[53,66], recent genetic analyses have found certain PRE–PRE *cis*-contacts (at other *Drosophila* genomic loci) contribute to more robust silencing[53,56,57,66]. Furthermore, we found physical proximity of linearly distal elements associated with repression tended to be cooperative, in contrast to the enhancer–promoter proximity associated with activation. This is consistent with a model in which individually rare, long-range contacts cooperatively reinforce a silent state.

Collectively, these observations have broad implications for our understanding of the links between chromatin structure and gene regulation and the approaches we use to study them. Given the distributed nature of the regulation uncovered by our analysis, we suggest that the existing reliance on pairwise methods, such as proximity ligation used in 3C approaches and pairwise analyses common in microscopy approaches, have provided a skewed view of transcriptional regulation which over-emphasizes enhancer–promoter contact. Supporting such a view, increasing genetic evidence has advocated for this more distributed, integrated view of transcriptional regulation in which redundancy is the norm, and even proximal elements may have separable, redundant functions that contribute to the robustness of expression[69–75]. Similarly, consistent with CNN's emphasis on structures predictive of the repressive state, recent data also support a major role for developmentally regulated repression that extends beyond simple enhancer decommissioning to expand our view of regulation beyond this focus on individual enhancer–promoter interactions[56,57,68].

The development of high-resolution, high-throughput, multiplexed imaging methods now provides an excellent opportunity to test hypotheses and models of more-complex interactions between genome folding and transcriptional regulation. Although the availability of these data sets is currently sparse[33,34], we expect to see considerable growth in these types of data in the near future and an increasing demand for computational approaches to leverage these data sets. Given the depth and complexity of these data sets, we expect machine and deep learning approaches will play an increasingly valuable role in their interpretation.

## Methods

### Statistical analyses

*P value calculation. Wilcoxon tests:* One-sample Wilcoxon signed-rank tests were used to compare the median of a population against an expected or standard value. The R function wilcox.test from the stats package was used for all one-sample Wilcoxon signed-rank tests. *P* values are calculated by normal approximation with the R function wilcox.test. Two-sample Wilcoxon rank-sum tests (equivalent to the Mann–Whitney test in the R implementation) were carried out to compare whether the two-sample distributions (not paired) have a median shift greater than the null hypothesis (parameter *mu*, default = 0). *P* values are calculated by normal approximation with the R function wilcox.test.

*Kolmogorov-Smirnov test (K-S test):* Two-sample K-S tests were carried out to test whether two samples were drawn from the same continuous distribution (null hypothesis), else they came from different distributions, using the empirical cumulative distribution functions of the two samples. The R function ks.test from the stats package was used to carry out all K-S tests.

*Polymer normalization.* 3D polymer coordinates were converted to pairwise distance matrices by calculating pairwise distances between all pairs of barcodes in order to remove the effect of stochastic rotation on the data. The data set was then split into train/validation/test sets with 60/20/20 proportions. All examples were then standardized to the training set mean and standard deviation.

Details of normalization can be found at:

https://github.com/aparna-arr/DeepLearningChromatinStructure/tree/master/CNN/KFoldXVal/KfoldXvalAbdA/src/ReadData.py

Missing data in individual examples was handled by applying linear interpolation to adjacent barcodes to estimate the position of the missing barcodes. Code for missing value imputation can be found at:

https://github.com/aparna-arr/DeepLearningChromatinStructure/tree/master/DataPreprocessing

### Simulations

*Molecular dynamics polymer simulation.* To perform simulations of 3D DNA structures with known RNA labels, the openmm[76] package () was used to perform molecular dynamics simulations of polymers resembling DNA, as well as force calculation functions from the python library for simulating chromatin polymers developed by the Mirny lab[77]. Polymers were constructed from 52 bonded monomers with no self-attraction. Forces applied to these monomers included a density (to simulate nuclear constraint) and a repulsion force at close distances between monomers. In total, 500 individual, unique polymer simulations were run in a thermodynamic space for 100 timesteps of 100 simulation steps each. The polymer trace of each timestep was taken as a separate example, resulting in 50000 individual simulated examples. The simulated data set was then split into train/dev/test with proportions of 60/20/20.

Parameters for the energy functions in these simulations, as well as all other parameters set in the polymer simulations, are recorded at:

https://github.com/aparna-arr/DeepLearningChromatinStructure/tree/master/PolymerSimulation

*Binary contact hypothesis simulation.* Labels were generated for the binary contact hypothesis simulation by measuring the 3D distance between monomers 20 and 40 (designated enhancer and promoter) for each example. To match the class imbalance of the ORCA data, distances were ranked and the polymer examples with the top 30% smallest distances were assigned a label of ON. The remaining polymers were assigned a label of OFF. Each example had a threshold percentage chance of being assigned a random label with 50% probability (ON or OFF). If a polymer was not assigned a random label, it was then assigned as per the binary contact rules.

Preprocessing script for the binary contact simulation can be found at:

https://github.com/aparna-arr/DeepLearningChromatinStructure/tree/master/CNN/RandomPolymerControl/process_scripts/process_polymers_noise_binary_contact.py

*Global compaction state hypothesis simulation.* Labels were generated for the global compaction state hypothesis simulation by measuring all pairwise distances of all monomers and calculating the median distance. Median distances were then ranked, and polymer examples with the top 30% distances were assigned a label of ON. The remaining polymers were assigned a label of OFF. Each example had a threshold percentage chance of being assigned a random label with 50% probability (ON or OFF). If a polymer was not assigned a random label, it was then assigned as per the rules for the global compaction rules.

Preprocessing script for the compaction simulation can be found at:

https://github.com/aparna-arr/DeepLearningChromatinStructure/tree/master/CNN/RandomPolymerControl/process_scripts/process_polymers_noise_compaction.py

*Hierarchical simulations.* An additional, more-complex simulation was designed to test the capabilities of the interpretation methods to disentangle interacting and hierarchical rules. This simulation contained four elements: one promoter (P) at position 20, two enhancers (E1, E2) at positions 5 and 35, and finally one silencer (S) at position 45.

Two variations on this simulation were run: a non-cooperative simulation, and a cooperative simulation. Contact thresholds were set to result in a similar ON/OFF proportion as the real data.

*Non-cooperative:* Either enhancer could activate the promoter with equal probability (0.8) by being in contact. However, the silencer's presence close to the promoter silenced the gene with high probability (0.8) regardless of whether an enhancer was in contact or not.

Preprocessing script for the non-cooperative simulation can be found at:

https://github.com/aparna-arr/DeepLearningChromatinStructure/tree/master/CNN/RandomPolymerControl/process_scripts/process_polymers_multi_no_coop_move_S_in.py

*Cooperative:* Either enhancer can activate the promoter with equal low probability (0.1) by being in contact, however if both enhancers are simultaneously in contact, the promoter has a high probability of being activated (0.8). However, the silencer's presence close to the promoter silences the gene with a high probability (0.8) regardless of whether either or both enhancers were in contact or not.

Preprocessing script for the non-cooperative simulation can be found at:

https://github.com/aparna-arr/DeepLearningChromatinStructure/tree/master/CNN/RandomPolymerControl/process_scripts/process_polymers_multiway_move_S_in_v2.py

*Simulating effects of non-structural features.* In order to simulate the addition of non-structural regulation of gene expression state in the polymer simulations, structures were assigned to expression states based on a probabilistic model. Each example had a 50% (or for Supplementary Fig. 7: 25% and 75%, respectively) chance of being assigned a random label, and if an example was selected to be assigned a random label, it had equal chance of being assigned either ON or OFF. If a polymer was not assigned a random label, it was then assigned as per the rules for either the binary contact hypothesis or the global compaction state hypothesis.

### Machine learning models

*Random forest.* RF is an ensemble machine learning algorithm that utilizes the creation of many individual decision trees, each working on a subset of input features, and in the classification problem these individual trees then vote on the optimal classification for an example. RFs have been shown to rival neural network performance in Kaggle competitions.

The RF model was built using the scikit-learn ensemble package[78], specifically the RandomForestClassifier object. The input data were a flattened matrix of pairwise distances, as in the CNN but vectorized for input to the RF, and the output was either ON or OFF.

An optimal RF model was chosen using a hyperparameter search over the *abd-A* fit model, in parallel to the CNN hyperparameter search. Of all, 150 models were run, and the optimal RF model chosen by the validation set AUC (ROC). Code for the hyperparameter search of the RF model, and details of training, can be found at:

https://github.com/aparna-arr/DeepLearningChromatinStructure/tree/master/RandomForestBest

https://github.com/aparna-arr/DeepLearningChromatinStructure/tree/master/RandomForestParamSearchAbdA

Once the best set of hyperparameters was chosen, the optimal RF model was trained on data for each of the three genes of interest, and final results measured on the test set, which is held out entirely during both training and hyperparameter selection.

### Deep learning models

All deep learning models were built from the Tensorflow (v2.2)[79] and Keras (v2.3)[80] python (3.6–3.8) packages as the base. The normalized pairwise distance map input data set was split into train/dev/test with a 60/20/20 proportion. Test examples were not fed to the network until evaluation time, after hyperparameters had already been set.

*Dense neural networks.* DNN models were built with Tensorflow. Initialization, architecture construction, parameter ranges, activations, and classification were all implemented using standard methods. Exact architecture details and schematics of tested DNN architectures can be found in Supplementary Fig. 2.

Exact specification and detailed parameterization of the DNN implementations for each gene model can be found at:

https://github.com/aparna-arr/DeepLearningChromatinStructure/tree/master/BestFCNNXVal

*Convolutional neural networks.* CNN models were built with Keras. All CNN models consisted of alternating convolutional and max pool layers with batch normalization and ReLu activations (for convolutional layers). The final layer of each model consisted of a Dense fully connected layer with a sigmoid function activation for binary classification. Exact architecture details and schematics of tested CNN architectures can be found in Supplementary Fig. 2.

Exact specification and detailed parameterization of the DNN implementations for each gene model can be found at:

For gene models:

https://github.com/aparna-arr/DeepLearningChromatinStructure/tree/master/CNN/KFoldXVal

For simulation models:

https://github.com/aparna-arr/DeepLearningChromatinStructure/tree/master/CNN/RandomPolymerControl

*Regularization and optimizer.* The optimizer chosen for all deep learning models was a variant of the standard Adaptive Moment Estimation (Adam)[81]: the AdamW[82] optimizer, which adds a weight decay term to the weight update and helps to prevent spurious overfitting during training. The additional parameter of weight decay was optimized during the hyperparameter search, as were all hyperparameters, and results can be found in Supplementary Data 3.

*Evaluation metrics.* Because both the ON and OFF classes are important to this problem, the central metric to compare model performance was AUC (ROC). Precision, recall, and f1 score were also evaluated, and all metrics for the top-performing model can be found in Supplementary Fig. 4a. Confusion matrices showing counts of true positive, true negative, false positive, and false negative calls can be found in Supplementary Fig. 4b.

*Hyperparameter search.* A grid hyperparameter search was performed for the initial testing of fully connected and convolutional neural net single-gene models. Values within a reasonable range for each hyperparameter were tested at equal intervals, with all possible combinations tried. The best model was then chosen based on the optimal dev set AUC (ROC) score for the *abd-A* gene, and evaluated on the test set. The sorted table of the top 50 models and all associated hyperparameter values can be found at Supplementary Data 3. Schematics of architectures that appeared within the top 50 models of the hyperparameter search can be found at Supplementary Fig. 2.

All details of the hyperparameter search, including all hyperparameters tested and all values of these hyperparameters, can be found at:

https://github.com/aparna-arr/DeepLearningChromatinStructure/tree/master/OriginalHyperParamSearch

The best-performing model (architecture and tuned hyperparameters) was then used for all subsequent instances of model training. This optimal model was then trained individually for each gene and simulation data set. Tenfold cross-validation was performed on all final neural network models. Results of cross-validation can be found at Supplementary Data 2.

Details of cross-validation implementations can be found at:

For gene models:

https://github.com/aparna-arr/DeepLearningChromatinStructure/tree/master/CNN/KFoldXVal

For simulation models:

https://github.com/aparna-arr/DeepLearningChromatinStructure/tree/master/CNN/RandomPolymerControl

https://github.com/aparna-arr/DeepLearningChromatinStructure/tree/master/CNN/KFoldBlanking/KfoldXvalBinContactSim_121019

https://github.com/aparna-arr/DeepLearningChromatinStructure/tree/master/CNN/KFoldBlanking/KfoldXvalCompactionSim_121019

*Data sufficiency analysis.* Data sufficiency analysis was performed for the best *abd-A* CNN model to examine whether the number of examples within the ORCA data set was indeed sufficient for robust neural network training. Subsets of data were removed from the training set, and the AUC (ROC) was calculated after training and testing the model on the reduced data set. To remain comparable, all AUC (ROC) values were calculated on the same dev set, which was held out from training of all models. Results of data sufficiency analysis can be found at Supplementary Fig. 3.

Details of data sufficiency analysis can be found at:

https://github.com/aparna-arr/DeepLearningChromatinStructure/tree/master/CNN/KFoldXVal/KfoldXvalAbdA/src/ModelDriver.py

*Robustness to experimental noise.* The robustness of the training algorithm to experimental noise (Fig. 1f) was assessed by adding random jitter to all points along with the measured polymer for all measured data prior to training.

**Model interpretation.** Model interpretation and analysis of barcode importance to final model prediction were assessed with two methods for all CNN models.

*Blanking analysis.* Barcodes in windows of a specific size in the test set pairwise distance matrices were "blanked", or set to normalized data set average, and the obfuscated data was passed through the fully trained model and an AUC (ROC) was calculated. This AUC was then compared with the base AUC for the test set with all data present. The window was then shifted with a step size of 1 and the process repeated. Additional window widths were tested from 1 to 30. For the multi-class model, the AUC (ROC) for each class was calculated. This simulated a

loss of data, as in image obscuring methods for CNNs[83–85]. By drop-in AUC score after obfuscation, the effect of those blanked barcodes' presence could be assessed empirically. These AUCs were then converted to a normalized predictability score, where 100 corresponded to AUC equal to the case without any data blanking, and 0 represented random performance. Error bounds were calculated by 10-fold resampling of train/test sets, and the standard error of the mean was calculated for each window over the normalized predictability scores of each of these test results.

Details of blanking analysis implementation for all models can be found at:

https://github.com/aparna-arr/DeepLearningChromatinStructure/tree/master/CNN/KFoldBlanking

*SHAP-based interpretability analysis.* SHAP[46], or SHapley Additive exPlanations, is a set of methods to explain the output of deep learning and machine learning models. The method GradientExplainer[46] was used to find the most important pixels in the individual test set examples for each possible class outcome. The top true positive images for each class were selected, and the SHAP values plotted after the data and model were analyzed by GradientExplainer with a reference set of the training set. These were then qualitatively analyzed for indications of class-specific important distance map patterns in the data.

Implementation of SHAP-based interpretability analysis and parameterization for GradientExplainer can be found at:

For gene models:

https://github.com/aparna-arr/DeepLearningChromatinStructure/tree/master/CNN/DeepGradientGenes

For simulation models:

https://github.com/aparna-arr/DeepLearningChromatinStructure/tree/master/CNN/RandomPolymerControl

*Top decile Aagregate SHAP analysis:* Aggregate ON SHAP maps was generated by plotting a new map of the average of the top decile of SHAP values obtained across any ON cell for each interaction in the matrix. Associated distance matrices were generated by plotting the median distance among interacting pairs from the top decile of SHAP values. In parallel, OFF aggregate SHAP maps were generated by plotting the average of the bottom decile of SHAP values obtained across all OFF cells for each interaction in the matrix, and associated median distances for OFF cells were also plotted.

*Cooperativity analysis:* Quantification of cooperativity was done by computing, for each pair of positions (a,b) with high SHAP values in example, the frequency of any other pair of positions (c,d) also having had high SHAP values in the same examples. High SHAP values were defined as being within the top percentile of SHAP values at that position across the population. These cooperativity measurements were then summarized in a map of average cooperativity by taking the average frequency of position (a,b), demonstrating cooperativity with any other interaction (c,d).

**Reporting summary.** Further information on research design is available in the Nature Research Reporting Summary linked to this article.

## Data availability

The data set used consists of 54,365 independent measurements of WT *Drosophila melanogaster* (8–12 hr) embryo single-cell 3D DNA traces (ORCA) paired with single-molecule intronic RNA FISH readouts for three genes (*Ubx, abd-A, Abd-B*), published in Mateo et al.[33]. Intronic RNA FISH intensities for all three genes were binarized as nascent transcripts detected (ON) or not detected (OFF). These data are available here: https://zenodo.org/record/4741214. Code for data set preprocessing can be found at: https://github.com/aparna-arr/DeepLearningChromatinStructure/tree/master/DataPreprocessing.

## Code availability

All code used for all analysis can be found at: https://github.com/aparna-arr/DeepLearningChromatinStructure. Code was written primarily in Python 3.8, and depended on methods from Numpy[86,87], Scikit-learn[78], and Matplotlib[88]. Additional supplementary code was written in MATLAB™ (R2019a) and R. The code is distributed under the MIT license.

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

## Acknowledgements

We thank Ansul Kundaje for critical feedback and discussions about machine learning. We thank members of the Boettiger lab for the critical reading of the manuscript. This work was supported in part by a Packard Foundation Fellowship to A.N.B., and NIH grants DGM132935A and U01 DK127419 (PI ANB). A.R. was supported by an NSF graduate research fellowship (DGE—1656518).

## Author contributions

A.R. conceived the project, performed the analysis, and interpreted the results, with advice and guidance from A.N.B. L.J.M., and S.E.M. assisted with interpretation of the results. A.R. and A.N.B. wrote the manuscript with critical input from L.J.M. and S.E.M.

## Competing interests

The authors declare no competing interests.
