## [Peer Review File · Nature Communications]

REVIEWERS' COMMENTS

Reviewer #1 (Remarks to the Author):

I think this is a very interesting manuscript, attacking a very important topic. In particular, its results on the link between 3D architecture and transcription could advance the current understanding of the faint correlation between gene-enhancer distance and activity state.

I list, briefly, a few main points I think require some critical clarification.

The deletion-inspired computational approach that removes sections of the test polymers prior to CNN prediction could be affected by conceptual problems. By resetting the distance of the blanked regions to the average value, in principle, the nature of the distance data is altered. For instance, the distance triangular inequality could become violated. Hence, the alteration of the structure of the considered data should be accounted for in the analyses. In fact, "blanking" is commonly used in computer vision because typically the different pixel of an image are "independent" one from the other. However, this is not the case for a distance matrix. So, the authors should provide evidences about such an issue and, in particular, its implication on their conclusions. Anyway, some statements (such as "these analyses indicate that much of the structural information the CNN has used to predict transcription activity is physically distributed across the domain") tend to remain qualitative and the analyses tools employed not sufficient to ground them in quantitative terms.

Similar considerations apply to the use of the SHAP approach, which is here employed as a black box to try to guess which 3D features are more predictive of transcription. How exactly SHAP works, why SHAP values are significant and statistically relevant, why it is "the tool" to be used here, etc. remains obscure. So, the SHAP based conclusions on the observed predictive features of patterns of domains of enhancer-promoter and enhancer-enhancer proximity remain largely qualitative, based on indicators whose significance is unexplored, and comparable with observations the reader can extract directly from the contact data themselves. The inferred connections to specific polymer models as representative of the state of chromatin are also extremely loose or just wrong (e.g., the authors state their chromatin picture is closer to "a polymer equilibrating in a confined volume" and refer specifically to the Fractal Globule model, which is not the prototypical example of a polymer equilibrating in a confined volume).

Summarising, while I find this manuscript interesting, focused on a very relevant topic and based on valuable experimental data, I am far from convinced about the appropriateness of the computational and statistical tools used in the analyses. Its conclusions are often qualitative and loose, and do not return a really novel picture of how 3D architecture is linked to transcription.

Reviewer #2 (Remarks to the Author):

This is a novel deep-learning approach to extracting 3D genome interaction network from their recent Super-resolution DNA imaging data, which could reveal more complex and subtle enhancer-enhancer, enhancer-silencer interactions beyond traditional enhancer-promoter ones. Together with simulations and in silico perturbations, in addition to compare with known animal mutants, the authors appear to be able to making important predictions on unconventional interaction modes and to challenging some previous enhancer-promoter centric study results. Overall the idea is clear, methodology and result are intriguing. It would help the

field to rethinking or reexamining some concepts and previous results under simpler models/inferences. Deep-learning, although has shown some remarkable to properties, also has its weakness. Before publication, I would like the authors (1) to compare their deep-learning method with at least some standard learning methods: e.g. SVM (Kernel based classification) and RandomForest (Tree-based regression), demonstrating the advantages/weakness of their CNN method under the current applications; (2) to indicate the effects of over-training (with respect to the “double descent” theory) and robustness (what happen to those complex/subtle interactions when noises were added), exactly what has been “memorized” by the deep-learning machine?

Reviewer #3 (Remarks to the Author):

The authors utilized the imaging data generated in their previous work (Mateo et al 2019) to study the link between the 3D structure of the bithorax gene cluster (BX- C) in *Drosophila* and the activity of the genes within this cluster. They trained a CNN on the relative physical distances between loci in the cluster to predict the ON/OFF state of each of the three individual genes Ubx, abd-A, and abd-B. The question asked is very topical, as it is not yet what exactly is the link between gene expression and chromosome 3D structure, particularly within contact domains. The authors then take an additional crucial step by looking "under the hood" of the trained CNN to understand what physical features are used by the model to predict gene activation. They conclude that this structural information is distributed across the domain and includes regions that had not been assigned before as regulatory elements for the individual genes, and that specific enhancer-promoter interactions do not play a major role in determining the ON/OFF state of any of the genes. This is a very innovative approach which applies state-of-the art machine learning methods to state-of-the art imaging data. I have only one major point (below), plus concern about a few claims that do not appear to be evident in the presented results and that will require some clarification.

Major point:

- Because there is no direct experimental validation for the paper's conclusions, the lack of statistics in the analysis of the CNN results weakens the strength of the conclusions. The authors make extensive use of the comparison between SHAP values and Distance to infer potential cooperativity between enhancers in modulating promoters. However, it is not completely clear how they make their calls, for example, for high versus low values. The authors need to introduce some statistical methodology to more systematically identify these regions and associations. For example, "...slightly elevated SHAP values extend downstream of the promoter to the intronic enhancers...", "There is some elevation of SHAP values around the genes that are not being predicted." What does "slightly", "some" exactly mean, how is it different from high? How do we set thresholds? Why have only certain regions in the Figure 4 maps been selected and not others? What criteria led to the selection of these regions' location and size? The way this is presented in the manuscript is quite arbitrary.

Other points:

- "...we compared different size blanking windows, from a single monomer (6 kb) up to 30 monomers (180 kb)."" This is the first time the concept of monomer is introduced into the text. Although it is explained in the Method section, it would help to briefly explain it here.

- " In both cases, only 50% of the population followed the model rule and the other 50% were randomly assigned to “ON” or “OFF”, in order to account for the stochastic/transient nature of TF binding and other such unmeasured, but essential interactions." The explanation for the

50% choice is not clear. The number seems pretty arbitrary. It would be important to test how varying the % that follows the model affects the performance of the CNN in the two models (i.e. E-P contact vs. compaction). Could that lead to some of the more complex patterns observed in the ORCA data?

- "Since the promoter in this simulation is nearer to the end of the polymer, fewer neighborhood monomers are available to infer the position of the promoter, and thus the effects of promoter blanking are greater. " I really do not see that in figure 2C, the drop at the promoter and at the enhancers are pretty much identical. If anything, the drop at the enhancer looks larger.

- " Also, the predictions were not always most sensitive at the ends of the domain...between the genes in regulating transcription." Simulations were done with a single fixed distance between P, E and domain boundary, which does not necessarily reflect the actual distribution of the BX-C genes, so the stated conclusions appear not completely supported. Additional simulations done by varying these parameters should be tested. (actually, figure 2i appears to be in general agreement with the compaction model in figure 2d).

- "On the left map, bright yellow at the coordinate representing the interaction between E1 and P and between E2 and P (white boxes) " White boxes are missing in fig 3d-e, making it difficult to follow the text and to link SHAP to Distance. Also, is there a way to combine SHAP and Distance into a single map that captures the relationship between the two (e.g. large pos. SHAP/low Dist., large neg SHAP/high Dist., etc.)?

- "For example, in the OFF cells, we find that it is the enhancer-silencer interaction that is marked as most predictive (Fig. 3e) and that accordingly, corresponds to proximity between the S and P positions in the distance map (Fig. 3e)" it might be the lack of the white box, but I do not see this effect so clearly in the figure.

- "To achieve this...as the regulatory structures behind the simulation data". This paragraph is quite confusing in terms of language and concepts, and is hard to follow (e.g. "...predictive it is of the ON state when it is at its most predictive", "To avoid bias from limiting sampling...",)

- "The corresponding distance map shows these SHAP values are important when small - indicative of proximity dependent regulation". What does this mean? Please clarify.

- Please add genomic distance axis in panels of Fig. 3 and 4 (maybe at the bottom?). It would make it easier to follow the description in the text. (e.g. "Sequences within ~50 kb of one another...", "Sequences 50-200 kb apart typically show...")

- It is not immediately clear why Distance in panels Fig 4b and Fig4d is in arbitrary units. Weren't simulations done so that they reflect overall distances in the actual data?

- "This surprising 'inverted' architecture suggests a solution for the paradox of separating enhancers and promoters while compacting the overall domain. Such unusual compaction also places significant constraints on the molecular mechanisms which achieve it." could a hierarchical looping organization of the domain fit within this model?

- "Applied to the independent enhancer example, we find no substantial correlation among any of the positions (Fig. 5b)". That does not seem to be that case in Fig5b. There is a region of high correlation in the ON example, around the silencer (S).

We thank the reviewers for their careful reading of our manuscript and numerous helpful suggestions for improving the analysis. We present here a line-by-line response to reviewer comments, describing the additions and revisions we have introduced to address these concerns.

Reviewer #1 (Remarks to the Author):

I think this is a very interesting manuscript, attacking a very important topic. In particular, its results on the link between 3D architecture and transcription could advance the current understanding of the faint correlation between gene-enhancer distance and activity state.

I list, briefly, a few main points I think require some critical clarification.

The deletion-inspired computational approach that removes sections of the test polymers prior to CNN prediction could be affected by conceptual problems. By resetting the distance of the blanked regions to the average value, in principle, the nature of the distance data is altered. For instance, the distance triangular inequality could become violated. Hence, the alteration of the structure of the considered data should be accounted for in the analyses. In fact, "blanking" is commonly used in computer vision because typically the different pixel of an image are "independent" one from the other. However, this is not the case for a distance matrix. So, the authors should provide evidences about such an issue and, in particular, its implication on their conclusions. Anyway, some statements (such as "these analyses indicate that much of the structural information the CNN has used to predict transcription activity is physically distributed across the domain") tend to remain qualitative and the analyses tools employed not sufficient to ground them in quantitative terms.

This helpful feedback identified a lack of clarity in our explanation of how blanking was performed. The blanking experiments did not alter the distance between the non-blanked regions, we just replaced these values with ones uncorrelated to either the ON or OFF state. We have added the following clarification to this section:

“Similar to genetic deletion strategies which test function by replacing candidate regions with neutral non-regulatory DNA to preserve genomic spacing, we “blanked” genomic windows of the polymers by replacing position data within the window with the dataset average value (**Fig. 2**). This removes any potentially informative information from this part of the polymer structure while preserving the pairwise distances among non-blanked points.”

We have also clarified qualitative statements throughout the manuscript, and added statistical tests for these comparisons as appropriate (see below). For example, the claim that the predictive information is “Physically distributed across the domain” has been clarified as follows:

“spanning 10s to 100s of kilobases of neighboring chromatin, where no individual element accounts for more than 20% of the total effect. Notably, while these predictive regions overlap domains identified by prior genetic analyses ^{36,43,44} they also extend beyond the most distal known enhancers of these genes.”

We appreciate the reviewers guidance in making this portion of the revised manuscript more precise.

Similar considerations apply to the use of the SHAP approach, which is here employed as a black box to try to guess which 3D features are more predictive of transcription. How exactly SHAP works, why SHAP values are significant and statistically relevant, why it is "the tool" to be used here, etc. remains obscure. So, the SHAP based conclusions on the observed predictive features of patterns of domains of enhancer-promoter and enhancer-enhancer proximity remain largely qualitative, based on indicators whose significance is unexplored, and comparable with observations the reader can extract directly from the contact data themselves. The inferred connections to specific polymer models as representative of the state of chromatin are also extremely loose or just wrong (e.g., the authors state their chromatin picture is closer to "a polymer equilibrating in a confined volume" and refer specifically to the Fractal Globule model, which is not the prototypical example of a polymer equilibrating in a confined volume).

On the use of SHAP

Our original manuscript provided only a brief introduction of SHAP, with reference only to the original work which introduced the method. While the approach has become widely used since first introduced in 2017, we recognize many readers will be unfamiliar with this still relatively recent tool. To better explain the approach and its use here, we have rewritten much of the section about the SHAP analysis, remade all of the previous SHAP related figures and added additional explanatory figures and supplemental figures on SHAP to add further explanations of how the methods worked. Please see revised **Figures 3-6, Supp. Figs. 8-12**, and the heavily revised text starting after the section heading "SHAP analysis identifies SHAP analysis identifies...". We highlight below some of these changes.

Importantly, SHAP analysis is an established method in the CNN literature, and we have directed the reader to the original work describing the approach in detail - Lundenberg and Lee 2017. Furthermore, we have added additional references (Lundberg et al. 2020; Adadi and Berrada 2018; Mi et al. 2020) which describe in more detail the use of SHAP, how it works, how it has been used, and how it compares with other methods.

In addition to these *a priori* justifications for using SHAP for interpretation of the CNN model, we believe the analysis presented here provides further validation of the utility of SHAP. In particular, this SHAP analysis identified a collection of structural properties of the data which were predictive for gene expression AND coincided with previous experimental results (which the CNN knew nothing about). For example, the association of the compact structure of the BX-C with its silent state. Also the physical association of particular upstream regulatory regions of Ubx with the Ubx promoter were identified as predictive of Ubx expression, even though the CNN was not given the information about which position was a promoter and which position was an enhancer. These examples clearly prove the utility and the ability of our SHAP analysis to extract biologically

interesting conclusions from the “black box” of the trained CNN. Notably, this analysis made *additional* predictions about regulatory interactions that have not been previously reported, such as the “inverted structure” we describe that is predictive of all the BX-C genes being repressed or the interaction between the PRE-containing region upstream of Abd-A and the Ubx gene as being predictive of repression of Ubx. We are testing several of these predictions currently in the lab.

Addressing the problematic qualitative language

Our original description relied overmuch on qualitative language, even though the model predictions are in fact all *quantitative*: (e.g. blanking enhancers with 5-monomer windows in the simulated data reduces performance to 85%, or blanking up to 2/5th of the domain in predicting abd-A expression for the structure data never reduced prediction performance below 40% -- all *quantitative* predictions as plotted in the figures. However, statistical analysis of the comparisons was also not presented. We have extensively revised these comparisons and added p-values throughout. For example:

“The performance reaches a minimum when the blanking window is centered on either enhancer (p=6.9e-08 for windows spanning the enhancer vs non-regulatory regions).”

“ All blanking windows cause a drop in predictability (for all but the few positions indicated, this was statistically significant p<0.05)”

“For example, the ~60 kb domain spanning the Ubx promoter and its cluster of upstream enhancers (box 1) shows significantly higher SHAP values (p<1e-20, K-S two-sample test) than a corresponding sized region (box 2) inside the gene body (Fig. 4a)”.

With respect to fractal globules

The comparison to the fractal globule is not a major point of the paper and has been removed in the current rewrite. As long unknotted polymers can only mix through reptation, they approach equilibrium very slowly and spend a long time equilibrating in a state that has been called a fractal globule (Lieberman-Aiden 2009, Mirny 2011, Halverson 2014, Mirny and Dekker 2016). Both the fractal globule and the equilibrium globule share the departure from the behavior of a regular lattice, typically drawn in in text-books and reviews to depict heterochromatin, in which linearly distal parts of the polymer are also the most distal in 3D. Our use of the word “equilibrating” was non-ideal, but is still distinct from ‘equilibrated’. In addition to removing the reference to “fractal globule” we have removed the word ‘equilibrating’ to avoid confusion with ‘equilibrated’.

Summarising, while I find this manuscript interesting, focused on a very relevant topic and based on valuable experimental data, I am far from convinced about the appropriateness of the computational and statistical tools used in the analyses. Its conclusions are often qualitative and loose, and do not return a really novel picture of how 3D architecture is linked to transcription.

Guided by the reviewer’s suggestions, the revised manuscript presents a much more thorough use of statistical tools and much clearer explanations of the computational approaches presented. The conclusions described in qualitative language are now supported by quantitative differences and measures of statistical significance.

We would argue that the overall conclusion about the predictive relationship between chromatin structure and gene expression which arises from these analyses provides timely support to an ongoing controversy and investigation about how 3D structure informs transcription. In particular, we believe that a picture in which pairwise enhancer-promoter contact takes a clear backseat to other structural features contrasts strongly with the text-book picture, frequently used as a premise in research articles as well, of the link between 3D architecture and transcription. Our analyses predict an important role for repressive contacts, contrasting an “enhancer”/ “activator” dominated view in the literature. They identify novel structural features such as the “inverted architecture” of the repressed BX-C. We believe the added statistical analyses illustrating the robustness of these conclusions and novel predictions will make them interesting to a broad audience of the genome structure and transcription communities.

We thank the reviewer for identifying the weaknesses in our original presentation and analysis and have heavily revised the analysis and quantification to address these concerns.

Reviewer #2 (Remarks to the Author):

This is a novel deep-learning approach to extracting 3D genome interaction network from their recent Super-resolution DNA imaging data, which could reveal more complex and subtle enhancer-enhancer, enhancer-silencer interactions beyond traditional enhancer-promoter ones. Together with simulations and in silico perturbations, in addition to compare with known animal mutants, the authors appear to be able to making important predictions on un-conventional interaction modes and to challenging some previous enhancer-promoter centric study results. Overall the idea is clear, methodology and result are intriguing. It would help the field to rethinking or reexamining some concepts and previous results under simpler models/inferences. Deep-learning, although has shown some remarkable to properties, also has its weakness. Before publication, I would like the authors (1) to compare their deep-learning method with at least some standard learning methods: e.g. SVM (Kernel based classification) and RandomForest (Tree-based regression), demonstrating the advantages/weakness of their CNN method under the current applications; (2) to indicate the effects of over-training (with respect to the “double descent” theory) and robustness (what happen to those complex/subtle interactions when noises were added), exactly what has been “memorized” by the deep-learning machine?

We are excited to see the reviewer found the work novel, the idea clear, and the results intriguing! We thank the reviewer for these helpful suggestions. We have greatly extended our original analysis to now include:

1. A comparison with Random Forest learning algorithms (see **Revised Fig. 1e**, **Revised Fig 1f**, and **New Supp. Table 2** and **New Supp. Table 4**)
 - While the Random Forest significantly outperformed the enhancer-promoter contact model, the CNN significantly outperformed the Random Forest ($p < 1e-4$ in a comparison of Odds Ratios and in comparison of AUC). For example the CNN's

performance for predicting Ubx expression was 70% higher in relative AUC (**Revised Fig 1e**).

- Given the difference in performance, the relatively limited contributions of structure to predicting expression for any model, the speed at which compact CNNs can be accurately trained (see below) and the relative complexity of the Random Forests, we prefer to focus the remaining discussion on the predictions of the CNN.
 - It is not our aim to provide a comprehensive comparison of all machine learning methods for this prediction problem. Our emphasis is on the identification of structural features which are predictive of transcription, and we are excited in follow up work to see if these predictive associations are causal by disrupting the predictive structures experimentally.
2. An analysis of training time, performance and cost (see **New Supp. Fig. 3** and **New Supp. Table 2**). We found that our model followed traditional behavior throughout training, and saw no evidence of a potential “double descent” improvement in performance (Nakkiran et al 2019).
 3. An analysis of algorithm performance when data is randomly removed (**New Supp. Fig 5**)
 4. An analysis of the robustness to added noise, for both the CNN approach and the Random Forest approach (see **Revised Fig. 1f**)
 5. An analysis of randomly shuffled data using the same training regime (See **New Supp. Fig. 4**). This analysis returned an AUC 0.5 - indicating no predictability. This confirms that there are not predictivity artefacts arising in the CNN training.

Analyses 2-5 demonstrate the robustness of the approach to missing data, noise, and training regime, and have substantially enhanced the paper. We thank the reviewer for these helpful suggestions. These new analyses are described in a substantially expanded section of the manuscript, see pages 4-7. The comparison with Random Forest is discussed in the new section: “Alternative machine learning approaches...”. The approaches used have been added to the description in the methods. We have added a further discussion about the limitations of the machine learning approach into our Discussion section, please see Discussion paragraph 2.

Reviewer #3 (Remarks to the Author):

The authors utilized the imaging data generated in their previous work (Mateo et al 2019) to study the link between the 3D structure of the bithorax gene cluster (BX- C) in *Drosophila* and the activity of the genes within this cluster. They trained a CNN on the relative physical distances between loci in the cluster to predict the ON/OFF state of each of the three individual genes Ubx, abd-A, and abd-B. The question asked is very topical, as it is not yet what exactly is the link between gene expression and chromosome 3D structure, particularly within contact domains. The authors then take an additional crucial step by looking "under the hood" of the trained CNN to understand what physical features are used by the model to predict gene activation. They conclude that this structural information is distributed across the domain and includes regions that had not been assigned before as regulatory elements for the individual genes, and that specific enhancer-promoter interactions do

not play a major role in determining the ON/OFF state of any of the genes. This is a very innovative approach which applies state-of-the art machine learning methods to state-of-the art imaging data. I have only one major point (below), plus concern about a few claims that do not appear to be evident in the presented results and that will require some clarification.

We are excited that the reviewer shares our excitement for the relevance of the question of how chromatin structure relates to gene expression and described the approach as innovative and state-of-the-art. We found the reviewer's critical reading and suggestions especially helpful in improving the manuscript, and have extensively revised the statistical treatment as discussed below.

Major point:

- Because there is no direct experimental validation for the paper's conclusions, the lack of statistics in the analysis of the CNN results weakens the strength of the conclusions. The authors make extensive use of the comparison between SHAP values and Distance to infer potential cooperativity between enhancers in modulating promoters. However, it is not completely clear how they make their calls, for example, for high versus low values. The authors need to introduce some statistical methodology to more systematically identify these regions and associations. For example, "...slightly elevated SHAP values extend downstream of the promoter to the intronic enhancers...", "There is some elevation of SHAP values around the genes that are not being predicted." What does "slightly", "some" exactly mean, how is it different from high? How do we set thresholds? Why have only certain regions in the Figure 4 maps been selected and not others? What criteria led to the selection of these regions' location and size? The way this is presented in the manuscript is quite arbitrary.

We agree completely that many of the comparisons lacked explanation of the statistical footing and that the rationale and criteria for selecting highlighted features was not presented. We have extensively revised the presentation of all the results. In particular:

- The quantitative differences in prediction upon blanking are presented on an intuitive scale of 0-100% predictability, and we have added p-values associated with all of the predictability comparisons discussed in the text.
- As the reviewer aptly identifies, the interpretation of "slightly elevated SHAP values" was much less clear and has been more extensively revised: For all positions in 52x52 map, we now compute a p-value that the SHAP value is significantly greater than 0 or significantly less than 0, see **Revised Figs 3-6**. A small number of points, especially those corresponding to the most linearly separated parts of the polymer (and thus the edges of the map) do not rise to statistical significance and the current data depth. However, this analysis shows that all regions of "elevated SHAP" discussed in the text are indeed statistically robustly distinct from 0 and contribute to statistically predicting the ON state. This additional analysis greatly strengthens the paper.
- Throughout the text, and the numerous comparisons of X increased, or Y is higher than Z, we have added appropriate statistical tests of the median or distribution and clearly

defined the test inline. We apologize for the omission and believe this statistical treatment strengthens the presentation of these observations.

The total number of potential interactions among the measured chromosome positions, $(52^{2/2})$, is significantly larger than would make for easy reading, so we selected by hand a few regions that we found interesting based on what we know about the underlying sequence elements. These examples included interactions between elements previously demonstrated experimentally to influence transcription, such as the interaction of the Ubx promoter and the enhancer dense region 30 kb to the right (upstream) of the promoter. We find this sort of example worth highlighting, since the CNN was not given any information about which positions in the structure correspond to promoters or enhancers, but it was able to learn this association. We added a white box on this region as a guide to the eye to help the reader follow along in the corresponding part of the figure, where the SHAP values are higher than in most of the rest of the map. *We have revised the text to make it clear this is just an example.* We also highlight examples of interactions for which we are not aware any experimental evidence has suggested they might influence transcription. These include enhancer-enhancer proximity dependent interactions predictive of expression of Ubx and interactions between the gene body of Ubx and the upstream region of abd-A, predictive of Ubx silencing. We single out these examples as they suggest interesting followup experiments and additional hypotheses about the mechanisms of how chromatin structure influences gene expression, which we expand upon in the discussion.

Some regions of the map were boxed and not discussed in the final version of our text, which we agree was confusing. These boxes have been removed. Other boxes were hard to see, and we have revised the figures to make this easier to see.

Other points:

- "...we compared different size blanking windows, from a single monomer (6 kb) up to 30 monomers (180 kb)."" This is the first time the concept of monomer is introduced into the text. Although it is explained in the Method section, it would help to briefly explain it here.

Revised:

"We explored the effect of blanking a single 6 kb step on the chromatin polymer (i.e. 1 monomer) up to 30 monomers (180 kb)"

- " In both cases, only 50% of the population followed the model rule and the other 50% were randomly assigned to "ON" or "OFF", in order to account for the stochastic/transient nature of TF binding and other such unmeasured, but essential interactions." The explanation for the 50% choice is not clear. The number seems pretty arbitrary. It would be important to test how varying the % that follows the model affects the performance of the CNN in the two models (i.e. E-P contact vs. compaction). Could that lead to some of the more complex patterns observed in the ORCA data?

This is a reasonable point. We have added a comparison of different cutoffs aside from 50% random. "To capture the effect that chromatin structure is not the sole determinant of transcriptional state, in both cases, only 50% of the population followed the model rule and the other 50% were randomly assigned to "ON" or "OFF". Analyses of simulations in which the contribution of structure was more or less predictive, with cut-offs at 25% and 75% of the population following the rule, showed different absolute predictivity changes as expected but similar results in the relative importance of each sequence (**Supp. Fig. 7**)."

We have also applied this analysis to the models used for adding the interpretation of the SHAP values, see Supp. Fig. 11, and associated text on page 13.

- "Since the promoter in this simulation is nearer to the end of the polymer, fewer neighborhood monomers are available to infer the position of the promoter, and thus the effects of promoter blanking are greater. " I really do not see that in figure 2C, the drop at the promoter and at the enhancers are pretty much identical. If anything, the drop at the enhancer looks larger.

This erroneous statement has been removed.

- " Also, the predictions were not always most sensitive at the ends of the domain...between the genes in regulating transcription." Simulations were done with a single fixed distance between P, E and domain boundary, which does not necessarily reflect the actual distribution of the BX-C genes, so the stated conclusions appear not completely supported. Additional simulations done by varying these parameters should be tested. (actually, figure 2i appears to be in general agreement with the compaction model in figure 2d).

Relating to the similarity between Fig 2i and 2d: Upon review we find the text was a bit unclear and has been modified -- we do see a correspondence between the data and the compaction model, and believe this does reflect an informative, major mechanism of repression. The point we wish to add is that compaction is not the only contribution in the data (unlike the model), as if it was, one would expect both ends to be equally sensitive, and one would not expect as many local minima in the domain. We have rewritten this to be clearer.

Additional simulations: We conducted additional simulations altering the E-P distances. At all scales, the analysis reliably pinpoints the E-P interaction. These simulations provide additional context to the reader for interpreting the main results, and are presented in the **New Supp. Fig. 10**.

- "On the left map, bright yellow at the coordinate representing the interaction between E1 and P and between E2 and P (white boxes) " White boxes are missing in fig 3d-e, making it difficult to follow the text and to link SHAP to Distance. Also, is there a way to combine SHAP and Distance into a single map that captures the relationship between the two (e.g. large pos. SHAP/low Dist., large neg SHAP/high Dist., etc.)?

We agree that bouncing between maps to find the corresponding positions is a little cumbersome. We explored combining these data in the same map, using 3D contour plots for 1 dataset (e.g.

distance) and color from the other dataset (e.g. SHAP). For maps with a small number of peaks, such as the simulations with 1 enhancer and 1 promoter, these maps are easier to read and immediately see the peaks match. However, with increasing amounts of structure in the data, we found the 3D-colored plots quickly become more difficult to read, especially when projected on a 2D page where the user can't rotate and explore them (see below).

Reviewer Fig 1:

For the simulation data:

For the abd-A "OFF" predictions

We hope that annotating corresponding regions that are discussed in the text with boxes improves the readability. The original manuscript omitted drawing some boxes which were discussed and contained some extra boxes which were not discussed, which detracted from their use. We have revised this. We have also adjusted the color of the boxes in some cases -- white squares do not show up well at all in Fig 3D for example.

- "For example, in the OFF cells, we find that it is the enhancer-silencer interaction that is marked as most predictive (Fig. 3e) and that accordingly, corresponds to proximity between the S and P positions in the distance map (Fig. 3e)" it might be the lack of the white box, but I do not see this effect so clearly in the figure.

We have remade this figure and added boxes, we believe the effect should be much more clearly visible in this revised presentation.

- "To achieve this...as the regulatory structures behind the simulation data". This paragraph is quite confusing in terms of language and concepts, and is hard to follow (e.g. "...predictive it is of the ON state when it is at is most predictive", "To avoid bias from limiting sampling...",)

Agreed. This section has been completely rewritten.

- "The corresponding distance map shows these SHAP values are important when small - indicative of proximity dependent regulation". What does this mean? Please clarify.

Agreed. This has been rewritten:

The corresponding position on the distance map shows this structural feature was predictive when its distances were small, <100 nm (**Fig. 4a**, box 1). This supports a contact/proximity dependent mechanism for enhancer function.

We have also expanded the description of how these population SHAP maps and population distance maps were generated with both additional text and additional figure panels, see discussion of Fig 3. We hope this combined explanation and rewrite makes this easier to understand.

- Please add genomic distance axis in panels of Fig. 3 and 4 (maybe at the bottom?). It would make it easier to follow the description in the text. (e.g. "Sequences within ~50 kb of one another...", "Sequences 50-200 kb apart typically show...")

The requested scale bars have been added to these figures. Thanks for the suggestion.

- It is not immediately clear why Distance in panels Fig 4b and Fig4d is in arbitrary units. Weren't simulations done so that they reflect overall distances in the actual data?

All distance plots using data have now been rendered in nanometers.

- "This surprising 'inverted' architecture suggests a solution for the paradox of separating enhancers and promoters while compacting the overall domain. Such unusual compaction also places significant constraints on the molecular mechanisms which achieve it." could a hierarchical looping organization of the domain fit within this model?

Some hierarchical looping models would be consistent with the 'inverted' architecture, though the simplest models that are consistent with it are not hierarchical. A carefully wrapped ball of yarn has an 'inverted structure' as at short length scales the strands cross the entire diameter of the ball, while at long length scales the strands frequently remeet one another.

We don't see a lot of bias about which distal points are closest, which is inconsistent with a hierarchical model in which there is a strong sequence preference for which elements fold at which stage in the hierarchy (i.e. if the model was 5' and 3' ends of genes loop first, and then exons within these regions loop together within these loops, we would see that structure in the data). A hierarchical looping in which the first level of loops had a strong preference for nearby interactions, and would also not be consistent. If higher order loops in the hierarchy only form *within* and not between lower order loops, it would be hard to achieve the high degree of separation.

- "Applied to the independent enhancer example, we find no substantial correlation among any of the positions (Fig. 5b)". That does not seem to be that case in Fig5b. There is a region of high correlation in the ON example, around the silencer (S).

We have developed an alternative approach for looking at higher order interactions which provides much clearer results on simulation data produced from models where higher-order regulatory interactions have been put in by hand. We have completely rewritten this section, and the new approach and new results are shown in the new **Figures 5 and 6**, and the corresponding **Supplementary Figure 12**.

REVIEWERS' COMMENTS

Reviewer #2 (Remarks to the Author):

The authors have substantially revised their manuscript based on the previous reviews. I think it is acceptable for publication in NCOMMS in its present form without further questions.

Reviewer #3 (Remarks to the Author):

This is a much improved version of the manuscript. The additional analyses and clarifications included in this revision address all the points I raised in my review satisfactorily.

Nicola Neretti

REVIEWERS' COMMENTS

Reviewer #2 (Remarks to the Author):

The authors have substantially revised their manuscript based on the previous reviews. I think it is acceptable for publication in NCOMMS in its present form without further questions.

Thank you!

Reviewer #3 (Remarks to the Author):

This is a much improved version of the manuscript. The additional analyses and clarifications included in this revision address all the points I raised in my review satisfactorily.

Nicola Neretti

Thank you!